# Machine learning models highlight environmental and genetic factors associated with the *Arabidopsis* circadian clock

Connor Reynolds[1], Joshua Colmer[1,2], Hannah Rees[3], Ehsan Khajouei[1], Rachel Rusholme-Pilcher[1], Hiroshi Kudoh [4], Antony N. Dodd [5] & Anthony Hall [1,6] ✉

The circadian clock of plants contributes to their survival and fitness. However, understanding clock function at the transcriptome level and its response to the environment requires assaying across high resolution time-course experiments. Generating these datasets is labour-intensive, costly and, in most cases, performed under tightly controlled laboratory conditions. To overcome these barriers, we have developed ChronoGauge: an ensemble model that can reliably estimate the endogenous circadian time of *Arabidopsis* plants using the expression of a handful of time-indicating genes within a single time-pointed transcriptomic sample. ChronoGauge can predict a plant's circadian time with high accuracy across unseen *Arabidopsis* bulk RNA-seq and microarray samples, and can be further applied to make non-random predictions across samples in non-model species, including field samples. Finally, we demonstrate how ChronoGauge can be applied to generate hypotheses regarding the response of the circadian transcriptome to specific genotypes or environmental conditions.

Like all eukaryotes, plants possess endogenous circadian clocks that align internal biological processes with the external day-and-night cycle. Across multiple groups of organisms, robust clocks that are correctly aligned with the day-night cycle have been associated with fitness advantages[1]. For plants, this includes increased survival, photosynthetic efficiency, and yield[2]. The clock has also been associated with the performance of many crops, and their adaptation to new latitudinal zones[3]. At the core of the circadian clock is a set of genes that form a transcriptional network of interlocking feedback loops which generate ~24-h rhythms[4]. These clock genes, in turn, regulate the rhythmic expression of thousands of genes that are thought to integrate much of the plant's biology with the 24-h cycle.

Assaying the plant circadian clock is fundamental to understanding its function and mechanism, but this is non-trivial as it requires high-resolution measurements across several days[5]. At the transcriptional level, investigating variation in circadian gene regulation is particularly challenging due to the costly and time-consuming protocols involved in generating these high-resolution time-course gene expression data under controlled conditions. Furthermore, there can be considerable variation between studies in the composition of the circadian-regulated transcriptome, with both biological and analytical factors underlying this[6,7].

These limitations can be partially overcome using mechanistic or machine learning (ML) models for predicting an organism's "circadian

[1]Earlham Institute, Norwich Research Park, Norwich, UK. [2]TraitSeq Ltd, Centrum, Norwich, UK. [3]Institute of Biological, Environmental & Rural Sciences (IBERS), Aberystwyth University, Aberystwyth, Ceredigion, UK. [4]Center for Ecological Research, Kyoto University, Otsu, Shiga, Japan. [5]John Innes Centre, Norwich Research Park, Norwich, UK. [6]School of Biological Sciences, University of East Anglia, Norwich, UK. ✉e-mail: anthony.hall@earlham.ac.uk

time" (CT) using the expression of genes as features (gene features, hereafter). The CT refers to the internal time of an organism irrespective of external cues, and is often used to label the time-of-sampling under free running conditions (continuous light (LL) in plants). This contrasts with the "zeitgeber time" (ZT), which refers to the experimental time relative to environmental cues, and is used to label the experimental time under a light-dark (LD) cycle. These predictive models are fit to labeled time-course gene expression data, which can then be used to estimate an organism's internal time in single time-pointed datasets. Time estimates in previous models trained on LL-condition data appear to correspond with both CT and ZT labels without using any mathematical adjustment[8–14], suggesting that circadian variation can be captured even when samples are harvested under a LD cycle. Thus, all time predictions here will hereafter be referred to as CT estimates. By comparing CT estimates with experimental time, it is possible to identify genotypic or external factors that impact clock function without obtaining a detailed, high-resolution time-series of data.

Several models have been developed for the estimation of CT in mammals. These include MolecularTimetable[8], ZeitZeiger[9], BIO_CLOCK[10], partial-least-squares-regression[11] (PLSR), TimeSignatR[12], TimeTeller[13] and Taufisher[14]. Except for MolecularTimetable[15], these methods have not been tested or benchmarked with plant transcriptome data. Additionally, ML models such as ZeitZeiger and TimeSignatR include only single predictors that identify a sparse set of time-indicating biomarkers that can include fewer than 50 genes. Feature selection is often important to reduce overfitting to noise in the training data, but it cannot fully account for systematic noise in gene expression in the form of batch effects across experimental groups[16]. These batch effects can reduce the reliability of individual models for CT estimation. Since at least a third of transcripts in *Arabidopsis* have evidence for circadian regulation[6], it is also unlikely that one small feature set could represent the full complexity of the circadian transcriptome and its integration with different biological pathways.

For a more comprehensive inference of CT from plant transcriptome data, we developed ChronoGauge—a ML bootstrap aggregating (bagging)-like ensemble model that can estimate the CT with high accuracy in *Arabidopsis* bulk RNA-seq and microarray samples. By identifying orthologs of gene features, the model can be applied to non-model plant species that lack the number of datasets and sample sizes required for training and evaluating a species-specific model. Unlike several other CT prediction models, ChronoGauge displays competitive performance when applied to single time-pointed samples without using within-study-normalization or batch correction procedures. We applied ChronoGauge to generate biological hypotheses on how genotype or environment affects circadian function. The ensemble approach of ChronoGauge also provides a "circadian fingerprint" for each sample, representing variation across individual sub-predictors. This can be interpreted to suggest specific biological pathways that are stable or dysregulated in response to experimental perturbations.

## Results

### ChronoGauge overview

ChronoGauge is a supervised ensemble regressor developed for plant RNA-seq data that can estimate the CT of a sample based on an input vector representing transcripts-per-kilobase-million (TPM) normalized gene expression. The model includes 100 neural-network (NN) sub-predictors that are each fit to a unique set of iteratively selected gene features, with CT estimates across sub-predictors being aggregated using a circular mean. Each NN sub-predictor is a relatively simple multi-layer-perceptron developed from a proof-of-concept model[17], which outputs the sine and cosine of the CT. Using a custom angular ($\theta$) loss function, the model is able to more appropriately converge across the 24-h (1440 min) modulus compared with conventional loss functions like mean-absolute-error (MAE) (Supplementary Fig. 1). The bagging-like ensemble improves the robustness of the model across

unseen data, whilst also providing a "circadian fingerprint" representing variation between sub-predictors within specific genotypes and experimental conditions. ChronoGauge was trained using *Arabidopsis* RNA-seq data, as this species has the largest number of suitable datasets for training and evaluating the model.

Generating the ensemble of sub-predictors requires an RNA-seq expression matrix with high time-point resolution for training. Including training samples from multiple experiments is also critical, as this reduces the extent to which ChronoGauge becomes overfit to specific datasets. In our analysis, we combined 4 separate time-course datasets from *Arabidopsis* (Supplementary Data 1). This included experiments under both LL conditions[18,19] (measured in CT) and LD cycles[19,20] (measured in ZT), giving a total of $N = 56$ samples. Replicates were averaged to ensure a balanced number of LL and LD samples (both are $N = 28$ after averaging). We note that these datasets included experiments entrained under LD cycles, including neutral-day (12:12) and long-day (16:8), but not short-day (8:16). These datasets were selected for training as they all included time-pointed samples that were derived from wild-type (WT) Col-0 whole seedlings harvested under relatively standard experimental conditions. We considered including a further LL time-course (*Takeoka* et al.[21]; $N = 35$) within the training set, but decided to exclude it after we found it had a significant batch effect with less robust rhythmic gene expression compared with other datasets (Supplementary Fig. 2). Time-pointed transcriptome experiments with contrasting genotypes, conditions or tissue types were not included in the training set, as we wanted to ensure these data were unseen when using ChronoGauge to generate hypotheses. Microarray datasets were not included for model training, as they are generally more prone to noise and have a smaller potential gene feature space compared with RNA-seq data.

We hypothesized that a feature set composed of highly rhythmic genes with diverse phases of expression would make intuitive biomarkers for CT prediction. However, the inclusion of LD samples in the training dataset meant that there was a risk of ChronoGauge being fit to gene features that were regulated by the LD cycle, but were not circadian-regulated. To reduce this risk, we applied the R package MetaCycle[22] using only the LL dataset by *Romanowski* et al.[18] to determine each gene's circadian rhythmicity Q-value and phase based on the meta2d approximation (described in Supplementary Data 2). Only genes with significant circadian rhythmicity ($Q < 0.05$) were used to inform feature selection (N genes = 13,474) (Fig. 1a). The *Romanowski* et al. dataset was specifically used here as it had the largest number of rhythmic genes compared with other LL time-course datasets (Supplementary Fig. 2b), thus we assumed it was most informative.

Gene feature sets for each of ChronoGauge's sub-predictors are generated using a custom sequential feature selection (SFS) wrapper method around the proposed NN architecture (Fig. 1b). This was developed to iteratively build unique sets of gene features with diverse expression phases using semi-randomised parameters. A shortlist of 150 genes was first selected using the top 25 rhythmic candidates across balanced phase groups (binned every 4 h). The gene sets are then constituted through a combined forward- and reverse selection approach using the fivefold cross-validation MAE of CT estimates as a cost function. The algorithm theoretically stops when the number of genes in the feature set reaches 40, but in practice, this value will not be reached in a scalable time-frame. We therefore only ran the algorithm within a specified duration of 6 computational hours, after which we identified the top-performing feature set across iterations. The ensemble is established by running the SFS multiple times, with a bootstrap first being applied to select 50% of genes at random from the feature space. This promotes feature set diversity across the ensemble (Fig. 1c). The optimal feature sets from each run were used to tune and train sub-predictors independently using fivefold cross-validation across the same training data. The trained sub-predictors are able to be applied to test data points, from which the CT estimates can be

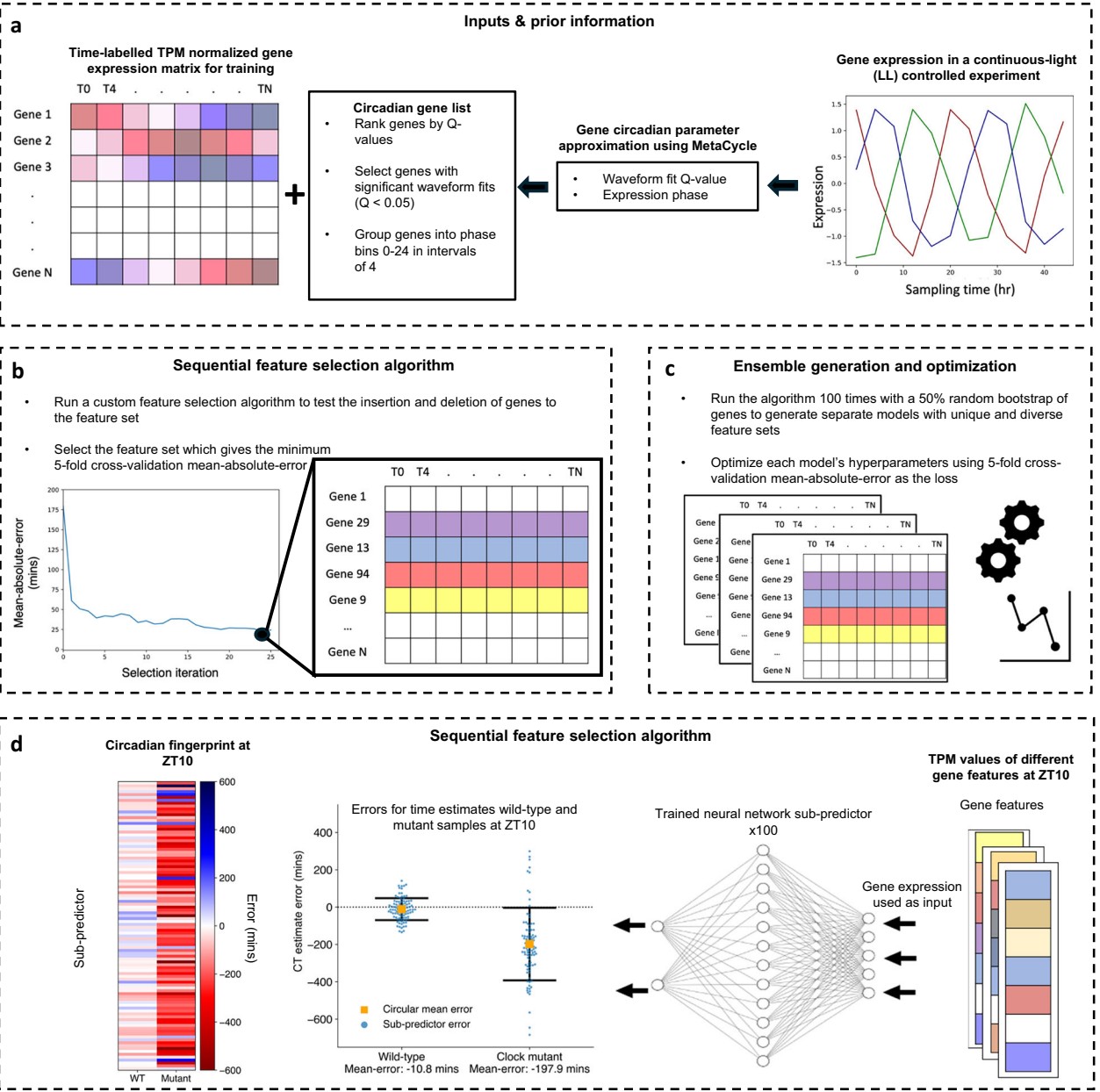

**Fig. 1 | Generation and application of ChronoGauge ensemble. a** Data input and prior information: ChronoGauge requires a gene expression matrix including observations from multiple time-course RNA-seq experiments for training and prior information regarding the circadian rhythmicity and phase of each gene's expression across a time-course harvested under continuous light (LL) following entrainment under a light-dark (LD) cycle as determined by MetaCycle. Genes with non-significant circadian rhythmicity (meta2d Q ≥ 0.05) are filtered out. Genes are grouped into 6 phase bins ranging 0−24 h in intervals of 4. **b** Feature selection: A custom sequential feature selection (SFS) algorithm involving both forward- and reverse-steps was used to iteratively build a gene feature set with diverse waveform phases, with the fivefold cross−validation (CV) mean-absolute-error (MAE) as a cost. The feature set giving the minimum MAE is selected from each run. **c**

**Ensemble generation and optimization:** The SFS algorithm is run 100 times with a random 50% bootstrap of genes, giving unique feature sets that are used to tune and train 100 different sub-predictor neural-networks (NNs). **d Single time point CT estimation:** Application of trained sub-predictors to single time-pointed test samples giving 100 different circadian time (CT) estimates that are combined into a single estimate using circular mean aggregation. The errors of CT estimates can be compared across different samples, e.g., between wild-type plants and clock mutants[28]. Within each sample, a circadian fingerprint represents variation in errors across sub-predictors fit to different gene features. Swarmplot properties include: central orange box = mean, whiskers = standard-deviation, blue points = errors for individual sub-predictors.

aggregated for each sample using their circular mean, or be analyzed as a "circadian fingerprint" (Fig. 1d).

## Evaluation of ChronoGauge ensemble's predictive accuracy

The SFS algorithm was run 100 times, giving 100 unique feature sets. Each set was used to train an individual sub-predictor within ChronoGauge's ensemble. The limit of 100 sub-predictors was set because the cumulative number of unique genes across new feature sets

appeared to plateau by 100 SFS runs (Fig. 2a). As part of model selection, we additionally evaluated two other NN-models using more biologically intuitive gene features that might be expected to be related to the circadian clock. This included a NN-model trained using 17 canonical plant circadian clock genes (genes were selected based on the fact they were used previously for circadian analyses[23] and were expressed in all training datasets, Supplementary Table 1), analogous to the mammalian model BIO_CLOCK[10], in addition to a NN-model fit to

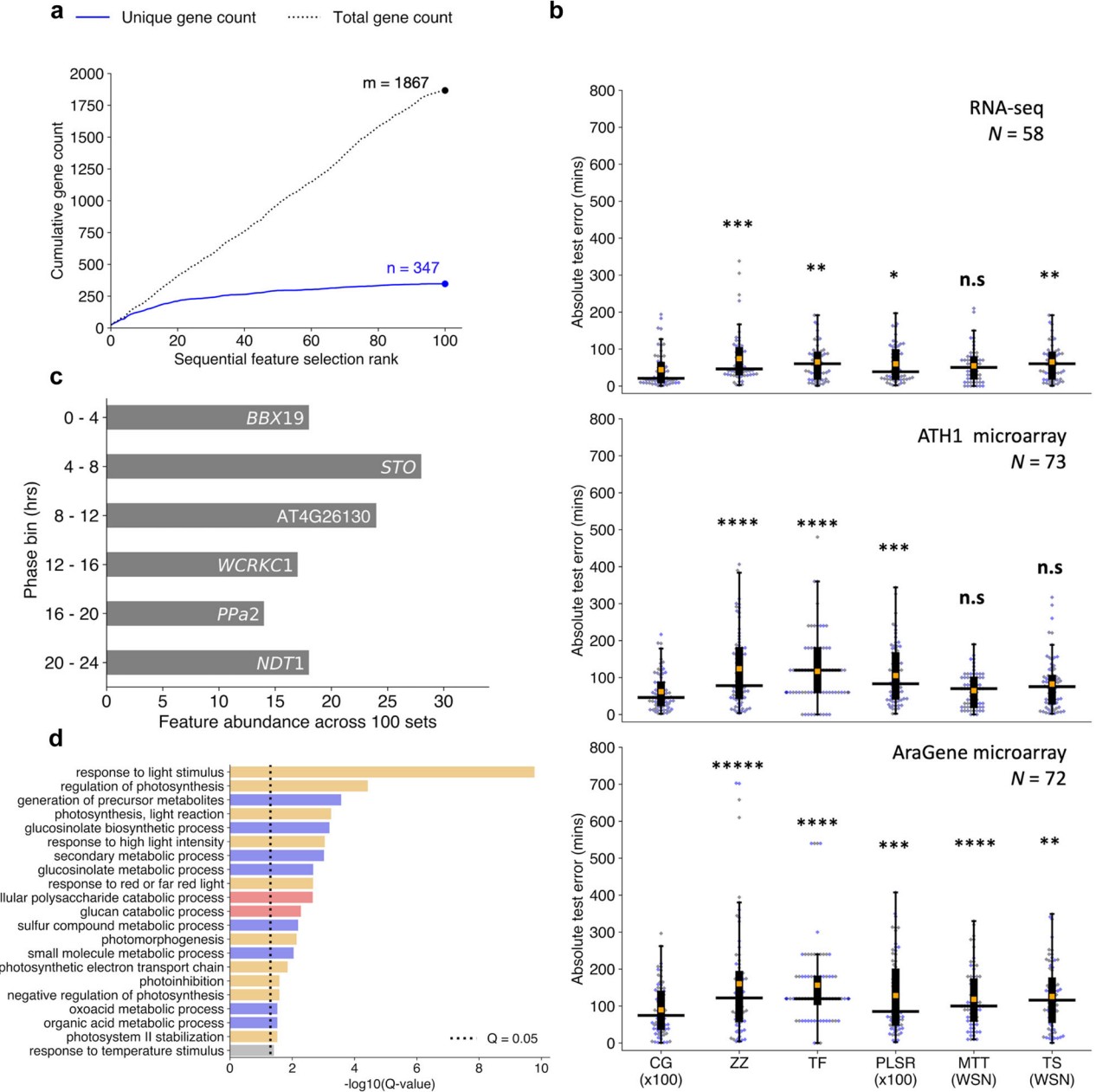

**Fig. 2 | Evaluation of ChronoGauge ensemble compared with other circadian time prediction models. a** Cumulative count of unique and non-unique genes present across sequentially selected feature sets ranked by fivefold cross-validation mean-absolute-error (MAE) of circadian time (CT) estimates. **b** Comparison of hold-out test set absolute errors between the ChronoGauge ensemble (CG ×100) and competing CT prediction methods including ZeitZeiger[9] (ZZ), Taufisher[14] (TF), partial-least-squares-regression ensemble[11] (PLSR ×100), MolecularTimetable[8] (MTT), and TimeSignatR[12] (TS). Comparison included for an RNA-seq test set[24–29] (N = 58), an ATH1 microarray set[30–33] (N = 73), and an AraGene microarray set[34] (N = 72). Bonferroni adjusted P-values shown from a one-tailed Wilcoxon signed-rank test. Methods using within-study-normalization (WSN) listed. Box-plot properties include: centre-line = median-absolute-error (MdAE), box limits = interquartile range (IQR), whiskers = 1.5 × IQR, orange box = mean-absolute-error (MAE), blue points = error of individual samples. **c** Top genes belonging to each phase bin based on the number of counts across the 100 feature sets within the ChronoGauge ensemble. **d** Significantly enriched biological process terms across the 347 unique selected gene features determined in TopGO[38] by Benjamni–Hochberg corrected Fisher's test using all circadian-regulated genes as a background. Processes include those related to light response/photosynthesis (orange), pest defense (blue), starch degradation (red), and temperature response (gray). n.s P ≥ 0.05, * P < 0.05, ** P < 0.01, *** P < 0.001, **** P < 0.0001, ***** P < 0.00001. Figure source data and actual P-values are provided as a Source Data file.

the top 500 rhythmic genes ranked by meta2d Q-values. The ensemble of 100 sub-predictors was ultimately selected as the optimal NN model for ChronoGauge, because it gave the lowest MAE across fivefold cross-validation (Supplementary Fig. 3).

We benchmarked the ChronoGauge ensemble across 3 unseen transcriptome datasets composed of WT and control *Arabidopsis* samples (including biological replicates) from multiple studies (Supplementary Data 3) to evaluate whether ChronoGauge's use was justified. Samples in these datasets were derived from either whole seedlings or aerial (including either leaf or shoot) tissue within an RNA-seq set[24–29] (N = 58), an ATH1-based microarray set[30–33] (N = 73), and an AraGene-based microarray set[34] (N = 72). While ChronoGauge was developed for RNA-seq data, we felt the inclusion of microarray data would enable an expanded evaluation of its generalizability and

**Table. 1 | Comparison of median-absolute-errors for hold-out test datasets across different circadian time prediction models**

| Model | RNA-seq | ATH1 microarray | AraGene microarray |
|---|---|---|---|
| ChronoGauge (×100) | **20.6 (± 47.6)** | **46.1 (± 50.2)** | **74.8 (± 68.1)** |
| ZeitZeitger | 46.2 (± 69.2) | 77.9 (±103.7) | 122.0 (± 156.6) |
| Taufisher | 90.0 (± 68.0) | 120.0 (± 93.6) | 120.0 (± 113.8) |
| PLSR (×100) | 38.4 (± 49.7) | 83.2 (± 77.6) | 85.5 (± 99.7) |
| MolecularTimetable | 50.0 (± 47.0) | 70.0 (± 44.3) | 100.0 (± 78.8) |
| TimeSignatR | 60.2 (± 49. 2) | 75.3 (± 67.9) | 100.4 (±105.0) |

Entries include the median-absolute-errors (MdAEs) for each circadian time estimation model across 3 hold-out test datasets. Bold text denotes smallest MdAEs within each test set. Paratheses denote the standard deviation of absolute errors across each test set.
*PLSR* partial-least-squares-regression, *×100* ensemble of 100 sub-predictors.

cross-platform application. We standardized the microarray datasets independently from the training RNA-seq using z-score scaling to allow the expression values to align across platforms. ChronoGauge was compared with other single-sample CT predictors, including ZeitZeiger[9], PLSR[11], and Taufisher[14] as well as predictors that rely on within-study-normalization, including MolecularTimetable[8] and TimeSignatutR[12]. Hyperparameters for these models (Supplementary Table 2) were tuned using a grid-search approach (Supplementary Fig. 4). PLSR was developed as a bagging-like ensemble of 100 sub-predictors using the same SFS algorithm described for ChronoGauge, since it was non-trivial to integrate this specific model within the wrapper. We did not include TimeTeller[13] because our training data was not suitable for input within this model.

All models gave CT estimates that were highly correlated (Pearson's correlation coefficient, $r > 0.9$) with the true sampling times (Supplementary Figs. 5–7). However, ChronoGauge gave the smallest median-absolute-errors (MdAEs) across each set at 20.6 (+/−47.6) min in the RNA-seq set, 46.1 (+/−50.2) min in the ATH1 set, and 74.8 (+/−68.0) min in the AraGene set (Table 1). These MdAEs were significantly smaller than the other models tested (one-tailed Wilcoxon signed-rank test, Adj. $P < 0.05$) in all cases except against MolecularTimetable in both the RNA-seq and ATH1 microarray set, and TimeSignatR in the ATH1 set (Fig. 2b), thus demonstrating ChronoGauge's general advantage in accuracy over other models across these hold-out test samples. MolecularTimetable was consistently placed in the top 3 models based on MdAE, which was unexpected considering it a mechanistic model that is substantially less complex than any of the ML methods. MolecularTimetable is limited however in that it requires the within-study-normalization of expression, which demands at least 2-timepoints ~12 h apart. This means it may not be suitable for predicting the CT within single time-point experiments.

In addition to using unadjusted expression data, we also tested each of the models after they were trained using RNA-seq expression data that had been adjusted by Combat-seq[35] to remove batch effects between experimental groups (Supplementary Tables 3–5). While intuitively, we would expect the removal of batch effects to improve accuracy by reducing technical derived variation, we found that ChronoGauge gave marginally (and non-significantly) smaller MdAEs when trained using unadjusted expression values for all test sets (Supplementary Figs. 8 and 9). The accuracy of ZeitZeiger, PLSR, and TimeSignaturR did give smaller MdAEs for RNA-seq data following batch correction, though none were competitive against ChronoGauge when using either adjusted or non-adjusted expression. Given that Combat-seq requires a laborious process of re-integrating different datasets and produced no performance advantage, we felt justified in using ChronoGauge with unadjusted gene expression as an input.

Our training data did not include a time-course experiment entrained to short-day conditions (only neutral- or long-day), thus it was unclear whether ChronoGauge could accurately estimate the CT with said short-day entrainment. Looking at model performance within each experiment (Supplementary Table 6), we do see that experiments

entrained and harvested under short-day conditions had higher MdAEs (though not unacceptable, using 120 min as a threshold) compared with all other conditions. This suggests overfitting has occurred, which on one hand may highlight differences in clock dynamics between plants entrained under short-day and neutral-/long-day photoperiods that have been observed previously[36]. On the other hand, it also suggests ChronoGauge may be less reliable when applied to samples harvested under short-day photoperiods. Despite the increased error, the fact that CT estimates are not random in the short-day experiments indicates that there is still some overlap in circadian gene expression compared with other experiments.

## Interpretability of ChronoGauge ensemble

Understanding which gene features are most contributory towards model prediction is useful both to validate that ChronoGauge is appropriately capturing circadian variation, and to potentially uncover mechanistic insights within circadian gene regulation. However, model interpretation using conventional feature importance methods such as SHAP[37] are less applicable for ChronoGauge with its ensemble of 100 sub-predictors. As ChronoGauge's SFS wrapper is discriminative in selecting genes, we instead explored the frequency in which genes were selected across the 100 runs within each phase bin. Except for *BBX19*, the most commonly selected genes in each phase bin did not include clock-associated components (Fig. 2c). Other clock genes, including *PRR7* and *RVE8*, were included in the top 5 genes of their respective phase bin (Supplementary Fig. 10). The top genes per phase bin displayed relatively consistent sinusoidal expression patterns across the training datasets with no obvious phase shifts across LL and LD experiments (Supplementary Fig. 11). While genes that were selected more often in SFS appear to be strong and logical predictors for inferring the CT, it is unclear whether they themselves give any insight into the clock mechanism or function.

To explore which biological pathways were enriched across all of ChronoGauge's 347 selected unique gene features from the 100 SFS runs, we performed GO-term analysis using all genes determined to be circadian-regulated (meta2d $Q$-value < 0.05) as a background (Fig. 2d). Since ChronoGauge's features tend to include highly rhythmic and robust genes, we believe significantly enriched processes (TopGO[38] Fisher's test, $Q < 0.05$) may represent pathways that are tightly regulated by the circadian clock. This included those related to light stimulus response and photosynthesis, which are pathways that are known to be particularly robust and conserved[23]. Multiple enriched processes were related to the synthesis of the defense secondary metabolite glucosinolate. While this defense pathway has previously been associated with the clock through both molecular and physiological observations[39], its enrichment suggests a particularly robust circadian regulation. Several processes were also associated with glucan catabolism, which is involved with the starch degradation pathway. Starch degradation is critical for providing energy for plant growth in the absence of light and is clock-regulated[40], thus its enrichment among our feature sets supports the suggestion that it is also a robustly regulated pathway.

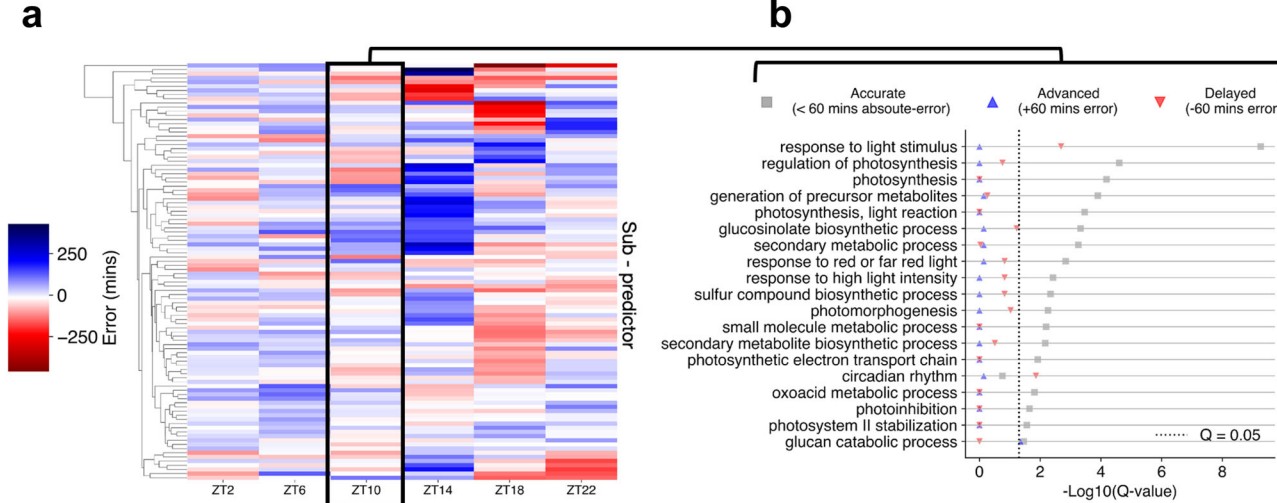

**Fig. 3 | Interpretation of sub-predictors across ChronoGauge's ensemble.**
**a** Comparison of phase estimates made by each sub-predictor within ChronoGauge for the test accession published by *Rugnone* et al.[24]. Colour-scale represents the mean-error of phase estimates made for each timepoint. **b** Enrichment of biological processes terms across different sub-predictor groups in ZT10, including those with accurate (<60 min absolute-error), advanced (+60 min error) and delayed (−60 min error) CT estimates. Term enrichment was determined using Fisher's exact test in TopGO[38] (*Q* < 0.05) using all circadian-regulated genes as a background. Significance determined by Benjamini−Hochberg adjusted *P*-values. Figure source data are provided as a Source Data file.

Irrespective of ChronoGauge's predictive performance lead over other models, an advantage of the ensemble is an increased ability to assess and dissect circadian function by comparing the gene features of groups of sub-predictors that give different CT estimates. To illustrate this, we demonstrated that there was substantial variation across sub-predictor outputs within each time-point of a 16:8 LD test dataset (Fig. 3a), including some that gave relatively accurate CT estimates and others that yielded advanced or delayed CT estimates. Using ZT10 as an example, we defined three groups of outputs across sub-predictors, including accurate (<60 min absolute error), advanced (> +60 min error) and delayed (< −60 min error), and performed GO term enrichment on each of these (Fig. 3b). We found that gene features within the accurate group were associated with most of the processes described using the full feature set in Fig. 2d. "Circadian rhythm" was the only process significantly enriched exclusively in the delayed group features, which included the clock components *PRR7*, *RVE8*, *BBX19*, and *CCA1*, as well as others associated with the clock. It is possible that some clock genes are not particularly reliable biomarker genes, as it has previously been shown clock gene expression values can exhibit considerable variation between plants at specific time-points[20]. This may also explain why our NN-based model using clock genes as features performed sub-optimally in cross-validation compared with the ensemble using multiple gene sets.

### ChronoGauge highlights dysfunction in core clock-associated genes

While ChronoGauge can make accurate CT estimates in wild-type (WT) control samples, we hypothesized that it could also detect dysfunction across the circadian transcriptional network in response to perturbations of clock components. To test this, we made CT estimates on RNA-seq datasets generated by *Graf* et al.[28] (Supplementary Data 4), which included the clock gene mutants *toc1-101* and *gi-201* in a Col-0 background, and *lhy-21/cca1-11* in a Ws-2 background. All samples included rosette tissue and were harvested under a 12:12 LD cycle. We compared the errors of CT estimates for each mutant with corresponding WT samples (Table 2) at both ZT0 (Fig. 4a–c) and ZT12 (Fig. 4d–f). Based on physiological assays of these mutants' period lengths, we anticipated they would display CT estimates that were advanced or delayed compared with the WT.

**Table 2 | Comparison of circadian time estimates made using across samples mutations of core circadian clock genes**

| Genotype | Background | ZT0 | ZT12 |
|---|---|---|---|
| Wild-type | Col-0 | −8.8 (±2.4) | 0.2 (±14.0) |
| *toc1-101* | Col-0 | 51.2 (±3.6) | 60.6 (±10.2) |
| *gi-201* | Col-0 | −12.1 (±7.6) | −96.9 (±7.2) |
| Wild-type | Ws-2 | 19.2 (±2.0) | 101.5 (±11.2) |
| *lhy-21/cca1-11* | Ws-2 | 70.4 (±13.6) | 388.9 (±2.2) |

Entries include the mean errors of circadian time estimates made by ChronoGauge for samples of clock gene mutants and time-point (*N* = 3) in Arabidopsis harvested under a 12:12 light-dark cycle. Paratheses denote the standard-deviation across each genotype and time-point. **ZT:** zeitgeber time.

In the Col-0 background (Fig. 4a, d), WT samples gave a mean-error of −8.8 (+/−2.4) min at ZT0 and 0.2 (+/−14.0) min at ZT12, corresponding to a period length of ~24 h (assuming an idealized 24-h clock period). *toc1-101* samples had significantly advanced CT estimates compared with the WT in both ZT0 and ZT12 with a mean-error of 51.2 (+/−3.6) min at ZT0 (Two-tailed independent *t*-test, Adj. *P* < 0.001) and 60.6 (+/−10.2) min at ZT12 (Adj. *P* = 0.02) based on a two-tailed independent-samples *T*-test. These advanced CT estimates correspond with a period length of ~23 h, which would be consistent with the previously observed phenotypes in *toc1* mutants, including an earlier phase under LD cycles[41] or a shorter period under LL[42,43]. The *gi-201* samples at ZT0 gave a mean-error of −12.1 (+/−7.6) min, with no significant difference found compared with the WT (Adj. *P* = 1.00). However, at ZT12, *gi-201* gave mean-errors of −96.9 (+/−7.2) min, suggesting a significant delay compared with the WT (Adj. *P* = 0.002). Previous studies have shown that *gi* mutants can display either long- or short-period phenotypes in LL depending on lighting conditions and the type of measurement taken across the assay[44,45]. The delayed CT estimate at ZT12 would correspond with a period length of ~25.5 h, indicating that these *gi* plants have a long-period phenotype. The relatively accurate CT estimate at ZT0 suggests that the entraining LD cycle may realign the clock in *gi* with the environmental cycle between dusk and dawn, with a misalignment emerging progressively through the photoperiod.

We performed GO term enrichment to find biological processes significantly associated (TopGO[38] Fisher's test, *Q* < 0.05) with gene

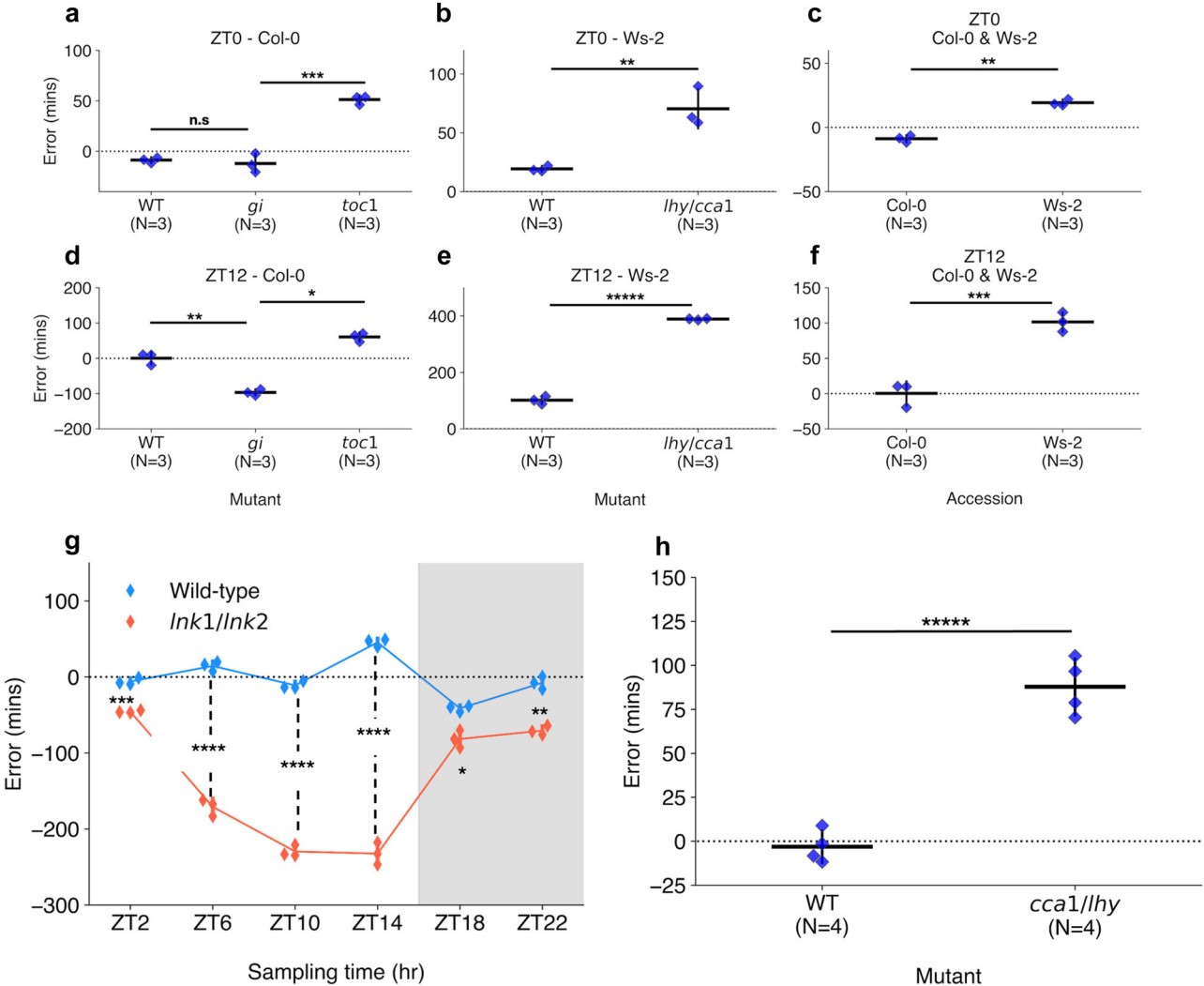

**Fig. 4 | Detection of perturbations to the core circadian clock.** Comparison of errors of circadian time (CT) estimates across clock mutant with wild-type samples (WT) and different accessions. CT estimates made across RNA-seq data generated by *Graf* et al.[28], including those harvested at (**a–c**) ZT0 and (**d–f**) ZT12 under a 12:12 light-dark (LD) cycle. Mutants include (**a, d**) *toc1-101* and *gi-201* in the Col-0 background and (**b, e**) lhy-*21/cca1-11* in the Ws-2 background. Additionally, (**c, f**) a comparison was made between Col-0 and Ws-2 accession CT estimates. **g** CT estimates across RNA-seq samples published by *Rugnone* et al.[24], including comparison of mean CT estimate errors in *lnk1/lnk2* mutant (red) and WT (blue) samples harvested across a time-course under a 16:8 LD cycle. White area shows

samples harvested under light, while gray area shows samples under dark conditions. **h** CT estimates across RNA-seq samples generated by *Blair* et al.[50], including comparison of CT estimates in *cca1-1/lhy-22* double mutant and WT samples harvested at ZT1 under a 12:12 LD cycle. All strip plot properties include: centre line = mean, whiskers = standard-deviation, blue points = individual samples. All tests performed using two-tailed independent *T*-test, with Bonferroni *P*-value adjustment for multiple tests within each experiment. n.s $P \geq 0.05$, * $P < 0.05$, ** $P < 0.01$, *** $P < 0.001$, **** $P < 0.0001$, ***** $P < 0.00001$. Figure source data and actual *P*-values are provided as a Source Data file.

features in accurate (absolute error <60 min), delayed (error < −60 min) and advanced (error > 60 min) sub-predictors across the samples under a Col-0 background (Supplementary Fig. 12). In *toc1-101*, GO-term processes including (but not limited to) circadian rhythm, temperature response, photomorphogenesis and red-light response were enriched in advanced error sub-predictors at both ZT0 and ZT12, suggesting a possible dysregulation to the circadian integration of light and temperature. In *gi-201*, most processes were significantly enriched in the accurate group at ZT0, suggesting only minor dysregulation. At ZT12 however, some enrichment was found across metabolic processes related to glucosinolate synthesis, suggesting a dysregulation of this pathway at this time-point. In both *toc1-101* and *gi−201* mutants, the glucan and polysaccharide catabolism processes associated with starch degradation were enriched only in the accurate groups. This suggests starch catabolism is a stably rhythmic diurnal pathway in LD, even under such clock perturbations.

In the Ws-2 background (Fig. 4b, e), WT samples gave a mean-error of 19.2 (+/−2.0) min at ZT0 and 101.5 (+/−11.2) min at ZT12. In the mutant samples, *lhy-21/cca1-11* at ZT0 gave a mean-error of 70.4 (+/−13.6) min, which was significantly advanced from the WT ($P < 0.01$), while ZT12 samples gave a mean-error of 388.9 (+/−2.2) min demonstrating a substantial advance ($P < 0.001$). Based on the difference at ZT12, we suggest the clock is dysregulated, which is consistent with the damped oscillations observed in previous *lhy/cca1* double mutants under LL[46]. The more moderate difference at ZT0 however, indicates that clock function may be somewhat rescued by the LD cycle between dusk and dawn. Following GO term analysis in *lhy-21/cca1-11* samples (Supplementary Fig. 13), gene features under the advanced (error > 90 min) or delayed (error < −90 min) sub-predictors for ZT0 were enriched mainly in pathways related to light response and photosynthesis regulation. At ZT12, most processes described in Fig. 2d were enriched in the advanced error group with few sub-predictors

belonging to the accurate group. We therefore suggest there is a widespread dysregulation of gene expression in the *lhy-21/cca1-11* genotype under LD cycles, especially at ZT12. This is consistent with the role of both *CCA1* and *LHY* as transcriptional repressors, which target large sets of dusk peaking transcripts[47,48]. We note the Ws-2 samples were significantly advanced ($P < 0.01$) from Col-0 at each timepoint (Fig. 4c, f). These advanced estimates likely reflect differences in circadian transcriptional dynamics between the two accessions, since ChronoGauge was trained only using Col-0 and not Ws-2, and also coincides with previous analyses that have shown Ws-2 plants display a marginally shorter period length than Col-0 plants (by ~30 min) under LL[49].

We further applied ChronoGauge to samples with other circadian clock mutants. In a 16:8 LD time-course spanning 24 h of aerial tissue samples[24], we observed a significant delay in CT estimates (Two-tailed *T*-test, Adj. $P < 0.05$) in *lnk1-1/lnk2-1* double mutant samples compared with the WT at all time-points, with highly significant delays (Adj. $P < 0.0001$) between ZT0 and ZT14 (Fig. 4g). A substantial dysregulation of the circadian network would be expected within the lights-on time-points in said mutants, as the *LNK* genes participate in integrating light stimuli with the circadian clock. In another experiment involving plants entrained to a 12:12 LD cycle[50] (including whole seedling tissue), *cca1-1/lhy-20* mutant replicates harvested at ZT1 were significantly advanced relative to the WT ($P < 0.001$) (Fig. 4h).

We also tested ChronoGauge in datasets obtained from plants challenged with different temperature conditions (Supplementary Fig. 14). In one experiment[50], Col-0 plants entrained at 22 °C were shifted to 10 °C or 37 °C one hour prior to sampling whole seedling tissue. Compared with control plants (kept at 22 °C), plants shifted to 37 °C gave significantly advanced CT estimates (Two-tailed *T*-test, Adj. $P < 0.0001$), while those shifted to 10 °C displayed no significant difference (Adj. $P = 1.00$). This suggests that while the circadian transcriptome is resilient to sudden drops in temperature, a sudden shift to 37 °C induces significant transcriptional differences, possibly because 37 °C is a stress condition for *Arabidopsis*. This is supported by the fact that the authors of the experiment reported a larger number of differentially regulated genes between 22 °C and 37 °C samples ($N = 3369$) compared with 22 °C and 10 °C samples ($N = 426$). It is difficult to associate these results with findings from independent works however, since few studies have explored the clock's response to short-term cold or heat stress. A separate experiment[27] tested differences in Col-0 plants grown at 22 °C and 27 °C under a short-day (8:16) LD cycle (including whole seedling tissue). CT estimates showed a significantly higher MAE across time-points in 27 °C samples (Two-tailed Wilcoxon signed-rank test, $P = 0.03$). In previous work, a subtle difference in the period lengths of Col-0 between plants that experienced long-term exposure to 27 °C compared with 22 °C[49], which coincides with our findings that exposure to higher temperatures for long durations can impact clock dynamics.

**ChronoGauge detects circadian variation in non-model species**
The previous analyses show that ChronoGauge can detect circadian clock variation in *Arabidopsis thaliana* circadian mutants and plants grown under different experimental temperatures. However, we wanted to explore its versatility further by making similar analyses in non-model plant species. Because the number of samples that are suitable for training models specific to other species are minimal, we instead identified putative orthologs of ChronoGauge's selected *Arabidopsis* gene features within each non-model species. Genes with no directly mapping orthologs were removed, and each sub-predictor was re-trained. Where multiple orthologous genes mapped to one *Arabidopsis* gene feature, their expression values were averaged. These steps ensured the feature spaces were identical across species. We tested this approach by making validation CT estimates on RNA-seq datasets, firstly including *Arabidopsis halleri* samples harvested from

the wild[51,52] ($N = 383$), as this dataset represents a closely related species to *A. thaliana* with the potential for comparison across different seasonal conditions. We also tested CT estimation in more distantly related species that contain known clock gene orthologs with time-courses that were sampled under controlled LL conditions, including *Brassica rapa*[53] ($N = 48$), *Glycine max*[15] ($N = 36$), and *Triticum aestivum*[23] ($N = 36$) (Supplementary Data 5).

Compared to the *A. thaliana* test data with a MdAE of 20.6 (+/−47.6) min, other species produced less accurate CT estimates. *A. halleri* field samples gave a MdAE of 137.1 (+/−162.0) min, which could reflect substantial noise in expression as the samples come from natural environments. In other species, *B. rapa* gave a MdAE of 169.2 (+/−70.3) min, *T. aestivum* of 141.5 (+/−90.4) min, and *G. max of* 79.0 (+/−50.1) min (Fig. 5a). It is unclear why *G. max* displayed such high accuracy compared with other non-model species, especially considering *B. rapa* is a closer relative to *A. thaliana*. One explanation could be that like *A. thaliana*, *G. max* is a diploid, thus they may have more orthologous genes that uniquely map to one another. In contrast, the paleopolyploid *B. rapa* and the hexaploid *T. aestivum* tend to include multiple ortholog mappings to individual genes in *A. thaliana*. These duplicated ortholog mappings may add noise to ChronoGauge's gene features, thus reducing its accuracy. We also noted that CT estimates were advanced in *B. rapa* and delayed in *T. aestivum*, which could be due to the clock period lengths being shorter and longer respectively in comparison with *A. thaliana* and *G. max*. Examining the period lengths of genes determined to be circadian-regulated (meta2d $Q < 0.05$; Supplementary Fig. 15), we found *G. max* genes had a significantly longer period length compared with *A. thaliana* genes (Two-tailed Mann–Whitney *U* test, Adj. $P < 0.00001$), with no significant difference between *B. rapa* and *A. thaliana* gene period lengths (Adj. $P = 0.94$). Thus, these transcriptome datasets suggest that ChronoGauge could not capture period length differences between these species. All CT estimates for non-model species displayed a high correlation ($r > 0.85$) with the true sampling times (Supplementary Fig. 16). The fact ChronoGauge was able to non-randomly estimate the CT across species as divergent as a monocot (*T. aestivum*) after training the model using expression data from the dicot *A. thaliana* suggests a high conservation in the circadian transcriptome that can be exploited to explore CT variation in species that lack appropriate training datasets.

The *A. halleri* samples[51,52] were harvested from a natural plant population in Hyogo Prefecture, Japan, across different time-points of the day throughout the months of March (3), June (6), September (9), and December (12). There was a strong correlation between the true sampling time (ZT hr) and the predicted CT estimates ($r = 0.88$) (Fig. 5b), though the samples tended to give more accurate CT estimates around ZT12, which roughly corresponds to dusk. This may indicate a stronger synchronization of the circadian transcriptome at dusk transitions. ChronoGauge is fit to training data entrained under square-wave LD cycles, while the *A. halleri* field samples would have experienced a photoperiod gradient from day to night. Thus, the higher error outside of dusk might also be explained by mismatches in the shape of transcript waveforms between controlled versus natural light-dark conditions.

We wished to evaluate whether ChronoGauge could identify differences between the circadian transcriptomes of the *A. halleri* samples that were caused by specific seasonal environmental factors, such as changes in the photoperiod length and ambient temperature. For samples with associated weather meta-data ($N = 367$), we made CT estimates to compare errors across seasons (Fig. 5c and Table 3). We found that plants harvested in June (6) and September (9) (giving median errors of −147.7 (+/−164.9) and −99.8 (+/−110.6) min respectively) were significantly delayed (Two-tailed Mann–Whitney *U* test, Adj. $P < 0.0001$) compared to March (3) and December (12) (with a median errors of 38.8 (+/−180.3) and 81.4 (+/−342.1) min, respectively).

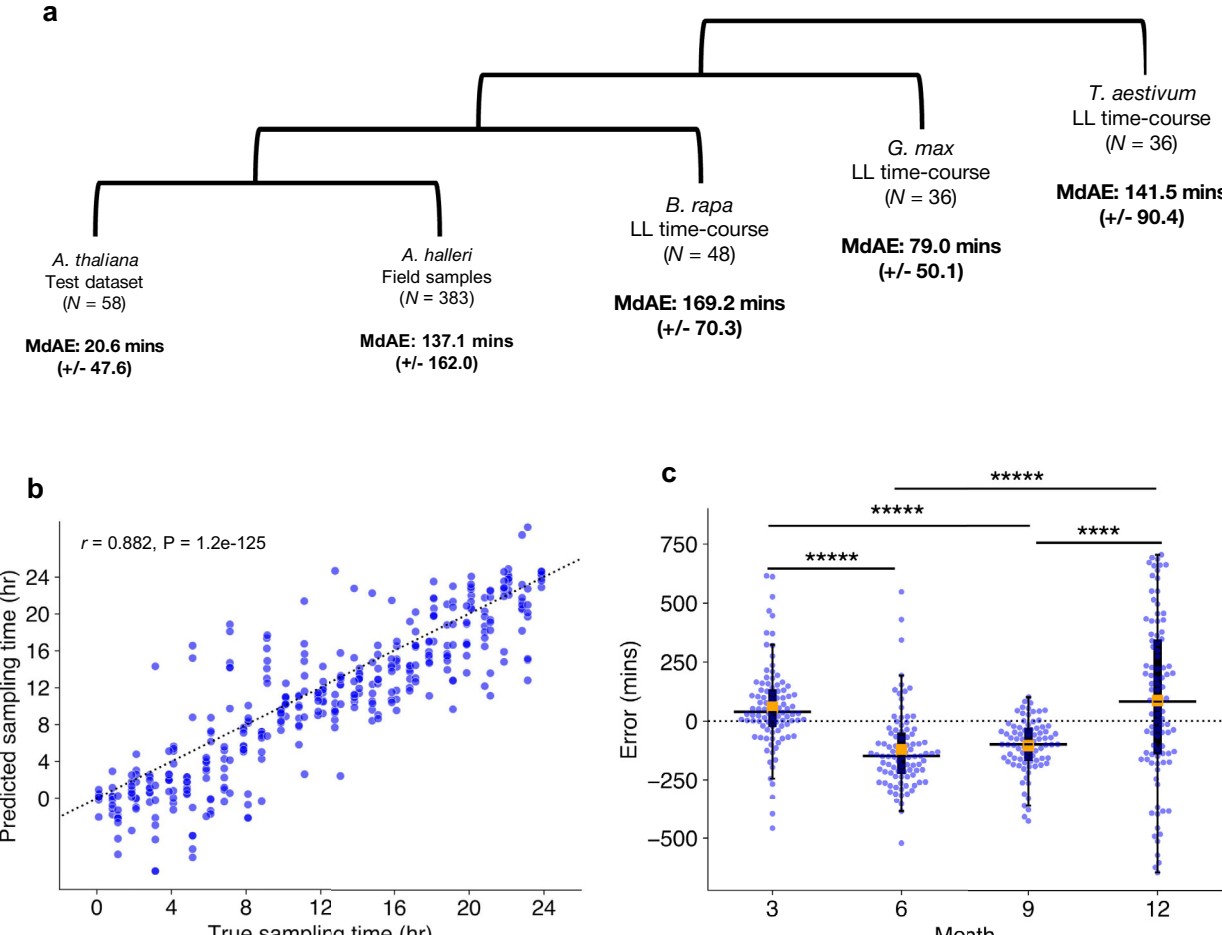

**Fig. 5 | Application to different plant species and environmental conditions.**
**a** Circadian time (CT) estimate mean-absolute-errors (MAEs) across test datasets in different species, the test dataset for *Arabidopsis thalina*, field samples for *Arabidopsis halleri*[51,52], a continuous light (LL) time-course for *Brassica rapa*[53], a LL time-course for *Glycine max*[15] (soybean), and another LL time-course for *Triticum aestivum*[23] (wheat). **b** Correlation between estimated sampling times/CTs with actual sampling times in *A. halleri* field data (N = 383) tested using Pearson's correlation coefficient. Sampling time labels were adjusted to set dawn to ZT0 based

on Tokyo time zone information. **c** Comparison of median CT estimate errors in samples with weather meta-data (N = 367) across different seasons, including Spring (3), Summer (6), Autumn (9), and Winter (12). Box-plot properties include: centre line = median, box limits = interquartile range (IQR), whiskers = 1.5 × IQR, orange box = mean, blue points = individual samples. Significant differences between group means determined using a two-tailed Man–Whitney U test with Bonferroni adjustment of P-values. **** Adj. P-value < 0.0001, ***** Adj. P-value < 0.00001. Figure source data and actual P-values are provided as a Source Data file.

On an absolute scale, CT estimates for winter samples gave the largest MdAE at 221.9 (+/−206.3) min (Supplementary Fig. 17), which was highly significantly greater (Adj. P < 0.001) than all other seasons, suggesting a less robust circadian regulation at this season. To test whether these differences might be explained by specific winter conditions, we associated sample errors with their environmental meta-data including photoperiod hours (sunset−sunrise), air temperature, precipitation and wind-speed (Supplementary Fig. 18). Air temperature and photoperiod, individually, displayed modest negative correlations with the CT errors (Pearson's correlation coefficient, $r < -0.30$), however, when adjusted for confounding variables using multivariate ordinary-least-squares regression, only air temperature was found to be significantly associated with error (multivariate P = 0.0009), while photoperiod was not (multivariate P = 0.13). Wind-speed and precipitation were not associated with CT error based either on individual correlation ($-0.05 > r < 0.05$) or when adjusted for multiple variables (multivariate P > 0.25). ChronoGauge thus suggests temperature may play a leading role in the regulation of the circadian clock across natural conditions and seasons, which is consistent with previous statistical analyses of these *A. halleri* field data[51].

**Table. 3 | Comparison of circadian time estimates made for *Arabidopsis halleri* wild samples harvested at different months under natural conditions**

| Month | Non-absolute | Absolute |
|---|---|---|
| March (N = 96) | 38.8 (±180.3) | 91.7 (±136.8) |
| June (N = 93) | −147.4 (±164.9) | 152.8 (±108.8) |
| September (N = 82) | −99.8 (±110.6) | 100.1 (±94.6) |
| December (N = 96) | 81.4 (±342.1) | 222.1 (±206.3) |
| All (N = 367) | −46.0 (±239.9) | 137.1 (±159.3) |

Entries include both the non-absolute and absolute median errors of circadian time (CT) estimates made by ChronoGauge for *Arabidopsis halleri* samples harvested from natural conditions at different months. Parentheses denote the standard-deviation across each month.

## ChronoGauge predictions highlight potential genetic marker-trait-associations
Natural variation in circadian rhythms has previously been observed across *Arabidopsis* accessions using leaf movement assays[49], luciferase imaging[54], and delayed-fluorescence[55]. Using ChronoGauge, we generated CT estimations from RNA-seq samples (rosette tissue) of 159

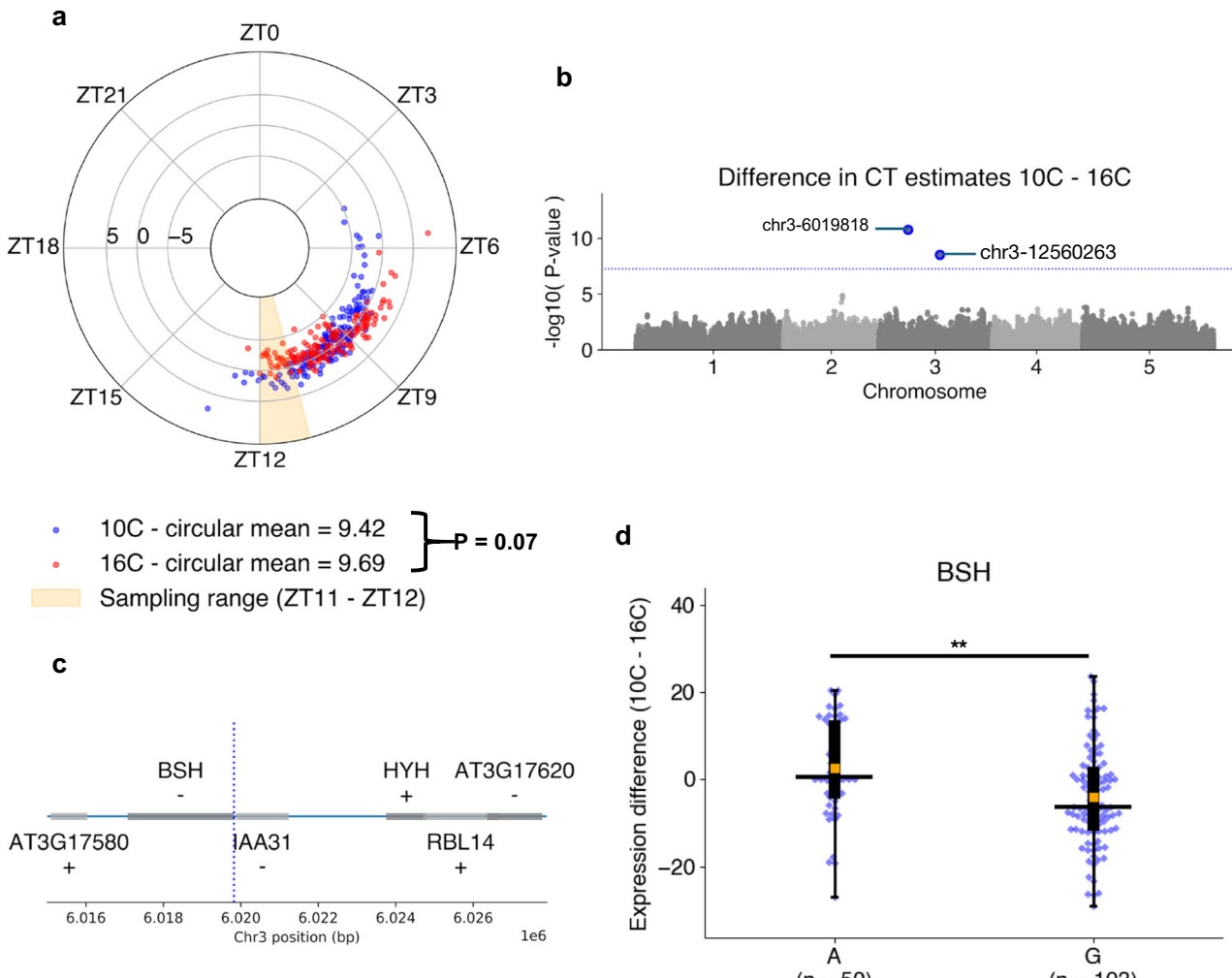

**Fig. 6 | Genome-wide-association-study identifies genomic marker sites associated with circadian response to cold. a** Circadian time (CT) estimates across 159 Swedish *Arabidopsis* accessions[56] grown under 10 °C (blue) or 16 °C (red) conditions. Polar axis determines ZT within a 24-h period. Radial axis represents the difference between temperature groups (10 °C–16 °C) for each ecotype. Statistical comparison of means between temperature groups performed using a paired *T*-test. **b** Associations between differential alleles and the difference between CT estimates 10 °C–16 °C across *N* = 153 accessions determined by the genome-wide-association-study (GWAS) model BLINK[78] in GAPIT[77]. Significant marker-trait-association (MTA) sites were identified with a threshold of *P* < 5.68e-08 due to Bonferroni adjustment across *N* = 880,417 SNP sites. **c** Genes proximal to the chr3-6019818 A > G MTA (blue line) identified through GWAS. **d** Comparison of transcript-per-kilobase-million (TPM) normalized expression differences of the *BSH* gene between 10 °C–16 °C groups in those containing the A and G allele at chr3-6019818. Box-plot properties include: centre-line = median, box limits = interquartile range (IQR), whiskers = 1.5 × IQR, orange box = mean, blue points = individual accessions. Statistical significance determined using two-tailed Mann–Whitney *U* test with Bonferroni adjusted *P*-values for all expressed and annotated genes 30 kb up- and down-stream the MTA site. ** Adj. *P*-value < 0.01. Figure source data and actual *P*-values are provided as a Source Data file except for Manhattan plots, which are provided within an Open Science Framework (OSF) file[79].

Swedish Arabidopsis accessions[56] to investigate variation in clock function (Fig. 6a). Each accession was grown at both 10 °C and 16 °C under 12:12 LD before being harvested for RNA-seq between ZT11 and ZT12. ChronoGauge gave a mean CT of 9.4 h (+/−1.5) in the 10 °C group and 9.7 h (+/−1.3) at 16 °C with a difference that was statistically modest (Paired samples *t*-test, *P* = 0.07). In comparison with previous phase approximations using delayed-fluorescence across a subset of accessions[55] (*N* = 20) (Supplementary Fig. 19), our CT estimates displayed a moderate correlation at 10 °C (Pearson correlation coefficient *r* = 0.50), but poor correlation at 16 °C (*r* = 0.07), suggesting ChronoGauge's transcriptome-based CT estimate is not necessarily consistent with other methods for quantifying clock parameters. The CT estimates for each temperature group were significantly earlier than ZT11 (the closest possible harvesting time based on mean estimates) (1-sample *T*-test, Adj. P < 0.001). The delayed CT estimates may indicate a long period phenotype for these ecotypes due to the lower growth temperatures

compared with the training samples (22–24 °C). This is consistent with previous literature showing long-term exposure to lower temperatures are associated with longer period lengths in many *Arabidopsis* ecotypes[49], though we note comparisons across independent experiments should not be over-interpreted due to the impact of batch effects.

We obtained variant allele information for 153 of these ecotypes, which included a total of *N* = 880,417 variant sites post-processing. We used three derived circadian phenotypes (CT estimates at 10 °C and 16 °C, and a temperature compensation phenotype calculated from the difference in CT estimates between the two temperatures 10 °C–16 °C) to perform a genome-wide-association-study (GWAS) using the BLINK model to identify marker-trait-associations (MTA) for each of the phenotypes. No significant MTAs were identified for the CT estimates themselves (Supplementary Fig. 20), but two significant MTAs were found in Chromosome 3 using the phenotype that was a proxy for temperature compensation of the clock (*P* < 5.7e-08) (Fig. 6b). For the

MTA at chr3-12560263, we found no annotated genes 30 kb up- or downstream. In comparison, the MTA at chr3-6019818 (Fig. 6c) lies within the promoter region of *BSH* (AT3G17590), a gene that has previously been associated with a "bushy" phenotype[57], but that lacks known association to the circadian clock or temperature responses. In gene expression data, the temperature response (TPM 10–16 °C) for *BSH* (Fig. 6d) was significantly greater (Two-tailed Mann−Whitney-*U* test, Adj. *P* = 0.008) across ecotypes with the A allele at chr3-6019818, 2.6 TPM (+/−11.1) compared to those with the G allele at −4.0 TPM (+/−11.2), supporting the notion that this gene is associated with a response to temperature of the Arabidopsis circadian clock. Interestingly, the gene *HYH* is ~4 kb from the MTA and has previously been shown to be implicated in a sigma factor-mediated cold response pathway[58] and regulation of a temperature-responsive miRNA[59], making it a second plausible candidate for a temperature response marker. However, we detected no *HYH* expression in any of the 153 samples, thus were unable to support its association with the temperature response phenotype using these data. Overall, these results demonstrate that ChronoGauge can be used to generate a circadian-related phenotype within a GWAS to identify potentially novel marker genes for clock regulation that are strong candidates for functional validation.

## Discussion

We have demonstrated that ChronoGauge's ensemble can estimate the circadian time of a plant transcriptome sample with outstanding accuracy within single time points without requiring the use of batch integration procedures across different datasets or within-study-normalization techniques. The NN-based architecture appears to be essential for this performance, as the linear-based PLSR model was significantly less accurate than ChronoGauge across test sets, despite both using an ensemble of 100 sub-predictors generated through an identical SFS framework. It is possible that the expression data collectively contains important and complex information that linear-based models are less able to exploit compared to a NN-transformation. While our ensemble of NNs may lack the intuition and interpretability of linear-based models, our results suggest a distinct advantage to its use for predicting complex phenotypic traits such as the CT. This coincides with previous research that demonstrates the use of non-linear ensemble models to reliably predict other phenotypic traits such as temperature stress[60] nitrogen use efficiency[61] from transcriptome data.

We suggest that the batch correction of gene counts through methods such as Combat-Seq may distort the complex information associated with the circadian transcriptome, since ChronoGauge displayed uncompetitive performance using the corrected expression values. In contrast, the linear-based models appeared to display improved performance following batch correction, likely due to the reduction in data complexity. We do note that mammalian circadian expression datasets often include samples from various tissue types with known transcriptional heterogeneities[62]. In these cases, a batch correction of expression data may be necessary to accurately estimate the CT. However, in the context of *Arabidopsis* bulk transcriptome samples, which tend to include whole seedlings or arial tissue, we saw no such requirement.

In addition to predicting perturbations to the circadian network, we demonstrated that the output from ChronoGauge can be interpreted to suggest circadian-regulated processes that are either stable or dysfunctional in response to environmental or genetic perturbations. While other CT predictors allow some interpretation of gene features and their contributions to the prediction, ChronoGauge's ensemble of models with multiple gene feature sets allows the identification of specific processes associated with accurate or inaccurate estimates. This gives us a "chronotype" for each mutant or environment treatment. We used this approach with clock mutant samples to show that processes related to starch degradation appeared stable irrespective of the perturbation (except for the *lhy-21/cca1-11* double

mutant), while those related to light response, photosynthesis, and pest defense can be become dysregulated even under LD conditions.

The ensemble included diverse feature sets, with some genes being selected in SFS more often than others. Without functional analyses, there are limited inferences that can be made about the top abundant genes, beyond that they make good CT predictors. While some of these genes are critical components of the circadian transcriptional network, the importance of other genes for the clock mechanism (if any) is unclear. Regardless of the biological reasons they are selected more frequently, ChronoGauge's predictive performance means that some of these genes could be potential markers for probing the circadian clock in more specific RNA-based experiments, such as spatial transcriptomics or RT-qPCR. However, further validation would be required to confirm the suitability of any combination of these genes as circadian markers in such contexts.

We found that ChronoGauge trained exclusively using *A. thaliana* expression data can display non-random CT estimates when applied to different plant species that have too few (*N* < 50) suitable samples to train and evaluate species-specific models. In particular, we were able to estimate the CT with a MdAE under 140 min across a large collection of *A. halleri* samples harvested from a natural population in the field, and could detect general differences to estimates made across different seasons and environmental conditions. While CT estimation error is higher than what was observed for *A. thaliana*, the fact that we observed non-random performances in non-model species with up to a 140-million-year evolutionary divergence indicates a high conservation between plant circadian gene regulatory networks that can be exploited by the model trained on *A. thaliana*.

We were also able to apply ChronoGauge to a large collection of *Arabidopsis* accessions, from which CT estimates can be used as a phenotypic trait in a GWAS model. Despite the population having limited variation in CT estimates, we identified a potentially interesting MTA at chr3-6019818 using the difference in CT estimates in response to alternate growth temperatures as a phenotype. This suggests the *BSH* gene may be associated with temperature regulation of the circadian clock because its promoter overlaps with the MTA site, and there is a significant difference in its expression response to growth temperature between allele groups. However, since no evidence currently exists to support an association between *BSH* and the circadian clock or temperature response, experimental analysis would be required to test this hypothesis.

A potential limitation of ChronoGauge is that it appeared to be overfit to data that included neutral-day or long-day entrainment, meaning applying ChronoGauge may give less reliable results when making comparisons of samples entrained under short-day conditions. This limitation could be overcome by including a high quality time-course spanning 2 or more days with short-day entrainment into the training data, but to the best of our knowledge, no such dataset is available.

In summary, we present ChronoGauge as the first CT estimation model developed specifically for plant transcriptome data. We identified that it has significant advantages in accuracy of CT estimation when applied to plants. ChronoGauge can be used as a diagnostic tool to assess plant clock dysfunction, and allows the interpretation of different biological processes that are stable or dysregulated. We show that it can be applied across multiple plant species, and gives outputs that can be used as a circadian phenotypic trait for GWAS models to identify MTAs. Together, our study demonstrates that ChronoGauge represents a powerful tool in both model and non-model plant species, to probe circadian clock-associated mechanisms and the impact of environmental factors on circadian regulation in both the lab and field.

## Methods

### Processing of RNA-seq datasets

Raw reads were downloaded for all accessions (Supplementary Data 1, 3–5) except the *A. halleri* dataset[51,52], which was provided pre-

processed at TPM level by the Kudoh Lab. Fastq files were processed first by removing adapter sequences using Trimmomatic[63] (v0.30) after which FastQC[64] (v0.11.4) was used to assess each sample's quality. HISAT2[65] (v2.0.4) was used to align reads to the reference genome of each accession's respective species, including *A. thaliana* TAIR10, *B. rapa* v1.0, *G. max* v2.1, and *T. aestivum* Chinese Spring RefSeq v1.1. Stringtie[66] (v1.3.3) was used to extract and quantify uniquely mapped reads at gene level. This included both raw read counts and TPM-normalized expression values.

## Processing of microarray datasets

Microarray accessions generated using the GeneChip ATH1 platform were downloaded in.CEL format, after which they were processed and normalized using the Robust Multiarray Averaging (RMA) function within the R package *affy*[67] (v1.80) in R v4.3.3. An accession generated using the Affymetrix GeneChip Arabidopsis Gene (AraGene) ST 1.0 platform was downloaded as a pre-processed expression matrix RMA normalized using the R package GC-RMA[68].

## Training dataset appraisal & prior circadian information

We initially selected 5 *A. thaliana* datasets[18–21] for model tuning and training (Supplementary Data 1). These were chosen as they were all RNA-seq time-course experiments that were time-point labeled, contained whole seedling tissue, and included relatively standard entertainment conditions. Datasets of specific tissue types or those that included experimental perturbations, including either the WT/control samples, were not included in the training data. The reason for this was because we wanted to use these datasets to test hypotheses, but this required both the WT and conditional samples to be unseen.

To evaluate the suitability of the proposed training data for modeling the circadian clock, we ran the R package MetaCycle[22] (v1.2) across each of the 5 datasets to obtain rhythmic parameters for each gene. Only genes that were expressed in each dataset were included. Circadian rhythmicity was determined using meta2d *Q*-values, which included the Benjamini–Hochberg adjusted ARSER and JTK_cycle *P*-values combined using Fisher's method. Genes were considered circadian if they were significantly rhythmic ($Q < 0.05$) in a LL experiment. After comparing the number of significantly rhythmic genes and general variation in gene expression across datasets, we chose to exclude the *Takeoka* et al.[21] dataset as it contained very few genes called as circadian-regulated compared with other LL experiments. Additionally, this dataset possessed a substantial batch effect relative to other experiments. The final training dataset consisted of $N = 56$ samples that included gene expression across 4 time-course experiments, with two being harvested under LL and another two under a LD cycle (both $N = 28$ when replicates were averaged).

To inform feature selection, prior knowledge of each gene's circadian rhythmicity and phase was determined exclusively using meta2d parameters from the *Romanowski* et al.[18] LL time-course (Supplementary Data 2). This was chosen because it included the largest number of rhythmic genes compared with any other dataset. Using the meta2d parameters, only genes with evidence for circadian rhythmicity ($Q < 0.05$) were used as potential features to prevent non-circadian genes being used as features. We additionally used the combined meta2d_phase metric as an approximation of each gene's circadian phase. Genes were binned into 6 different phase groups (phases 0–4, 4–8, 8–12, 12–16, 16–20, and 20–24) to enable a balanced selection of gene features with diverse waveforms.

## Benchmark datasets

We identified 6 Col-0 RNA-seq datasets for benchmarking the model, giving a total of $N = 58$ samples, including replicates. One dataset was a LL time-course including samples harvested from the shoot apex[24]. The other 5 test datasets[24,25,27–29] included WT and control samples (all whole seedling or leaf tissue) for experiments that also

included either experimental perturbations like temperature shifts and water deprivation, or different genotypes. We selected these datasets for model testing rather than training to ensure we could make an unseen comparison between WT/control and perturbation. TPM values for the training data were standardized using *z*-score scaling. All RNA-seq test datasets were fit to the scaling factor of the training data.

Microarray data was used only for model testing. They were not used for model training because the model was initially developed for bulk RNA-seq, and microarray data is considered more prone to noise with a smaller gene feature space compared with RNA-seq. One microarray test set included ATH1 platform test samples[30–33] ($N = 73$) normalized using *affy*[67] RMA, while another set included AraGene platform samples[34] ($N = 72$) that were acquired pre-processed and normalized using GC-RMA[68]. These were used in separate sets as their normalized expression values did not correspond with one another. We used *z*-score scaling to standardize expression values in each set independently of the training data. This independent scaling approach ensures the RNA-seq TPM expression in the training data corresponds with the microarray data in model testing.

## Batch correction using Combat-seq

To remove batch effects in gene expression values between experiments, we first applied Combat-seq[35] within the R package *sva*[69] (v3.50) on raw read counts of the training dataset, with each experimental group being used as a batch covariate. The adjusted counts were then TPM normalized. The training data's batch correction was completely naïve to the test samples to prevent data leakage.

For unseen RNA-seq test data, we merged the raw counts of test sample reads into the original raw training count matrix and applied Combat-seq. This was performed individually for each experiment to prevent data leakage across different test experiments. Each test dataset was extracted from the adjusted count matrix, discarding the training samples, before being normalized to TPM expression. We did not apply batch correction to microarray data as Combat-seq requires raw counts as an input, which are not applicable to microarray platforms.

## ChronoGauge neural-network

ChronoGauge's base model is an extension of a proof-of-concept NN model described previously[17] for CT estimation. The model (Supplementary Fig. 1) was developed using Tensorflow[70] (v2.6, using Python v3.9.5) as a multi-layer-perceptron NN that includes 3 fully-connected hidden-layers, with 2 output nodes including sine- (1) and cosine-transformed (2) values of the time-of-day within a 24-h modulus:

$$time_{\sin} = \sin\left(\frac{2\pi \times time_{ct}}{24} + \frac{\pi}{2}\right) \tag{1}$$

$$time_{\cos} = -\cos\left(\frac{2\pi \times time_{ct}}{24} + \frac{\pi}{2}\right) \tag{2}$$

Where $time_{ct}$ represents the hourly time-of-day within a 24-h modulus (e.g., a CT or ZT of 50-h becomes 2-h). A custom loss function was defined that uses $\Theta$ (3)—the angular distance between the sine and cosine values of the actual sampling times and the predicted times:

$$\Theta = \frac{\sum_{i=0}^{n} true_{time_{\sin} \times time_{-\cos}} \cdot pred_{time_{\sin} \times time_{-\cos}}}{\| true_{time_{\sin} \times time_{-\cos}} \| \cdot \| pred_{time_{\sin} \times time_{-\cos}} \|} \tag{3}$$

Where $true_{time}$ represents the actual harvesting times of the samples and $pred_{time}$ the CT estimates of the model in said samples. The cyclical outputs alongside the $\Theta$ loss function ensures the model appropriately considers the circular nature of the time-of-day during training. The sine and cosine outputs can be converted into the single

time-of-day value $time_{ct}$ (4) using an inverse tangent function:

$$time_{ct} = \frac{atan2(time_{\sin} . time_{-\cos}) \times 24}{2\pi} \qquad (4)$$

### Neural network-models using clock genes and top rhythmic gene features

Using the aforementioned NN architecture, we trained two models using biologically informed features. One was inspired by BIO_CLOCK[10], a similar NN-model that used mammalian clock genes as features. As the model is not currently accessible for re-training in *Arabidopsis*, we used our own NN-model, which was fit to the expression of 17 canonical clock genes hand-picked based on biological knowledge (Supplementary Table 1). These genes have been used to compare circadian rhythms in previous work[23]. Another was trained using the top 500 rhythmic genes approximated by the MetaCycle $Q$-value as features. Both models were tuned using a random search, using the fivefold cross-validation MAE as a cost.

### ChronoGauge sequential feature selection algorithm

The SFS algorithm was developed to semi-randomly generate unique and diverse feature sets for CT estimation using our NN-model. The algorithm initiates with a random bootstrap of 50% of gene features across those determined to be circadian-regulated (meta2d $Q < 0.05$). The top 25 genes from each phase bin are then selected, giving a range of 150 potential features from which one gene is selected at random to initialize the feature set. The algorithm then selects a phase bin that is underrepresented in the current feature set; if there are multiple underrepresented phase bins, then one will be selected at random. The algorithm iteratively tests inserting genes belonging to the selected phase bin one at a time to the feature set and reports the MAE across fivefold cross-validation. The gene whose insertion gave the minimum MAE is added to the feature set moving forward.

The iterative process of selecting underrepresented phase bins, then adding the gene which provides the greatest performance is repeated to build a feature set composed of rhythmic genes with diverse waveform phases. When the gene features $N = 8$, the algorithm also applies a reverse selection approach to remove genes from the feature set if it minimizes the cross-validation MAE. The reverse selection does not test removing the newest gene addition and will cease if the gene feature set is reduced to $N = 3$. The SFS algorithm as a whole ceases when the gene feature set $N = 40$, however, we note that in practice, reaching this number would take an unacceptable amount of time. We chose to use a time-limit of 6 CPU hours to run the algorithm, then selected the feature set which gave the lowest cross-validation MAE across all iterations.

### ChronoGauge ensemble generation & application

To generate the ensemble of sub-predictors, we ran the SFS algorithm using the NN model 100 times each for 6 computational hours on a non-GPU HPC node with one CPU (Intel Xenon Silver 4310 CPU @ 2.10 GHz) and 64 Gb RAM. After completion, we selected the most accurate feature set within each SFS run based on the cross-validation MAE. Using the selected feature sets, we optimized 100 sub-predictors using a random search for 12 computational hours (same HPC using 32 Gb RAM), each with cross-validation MAE as a cost, before finally training each sub-predictor using optimal hyperparameters. Hyperparameters in this random search included the learning rate, batch size, and l2 regularization factor. The sine and cosine outputs of the NN-models were transformed to a daily time 0–24 h and were aggregated using a circular mean.

We note that the genes selected using SFS may not be present in unseen data. For RNA-seq tests, missing gene features set to a TPM of 0, since it can be assumed these genes are very lowly expressed. For microarray datasets, however, several gene features are not present in

the ATH1 or AraGene probesets, and it cannot be assumed these genes are lowly expressed. We therefore chose to train ChronoGauge for each platform independently by removing any gene from the ensemble's feature sets that were not present in each of the probesets. Thus, three ChronoGauge models were trained that were specific to each of the platforms (RNA-seq, ATH1, and AraGene).

### Training and testing other CT estimation models

To evaluate ChronoGauge's ensemble across test datasets, previously published CT estimation models were trained and tested using the same datasets described previously. MolecularTimetable was implemented as a custom Python script as described by *Ueda* et al.[8] except for the Pearson correlation coefficient $r$ threshold, which was set to 0.89 as this gave a lowest MAE in the training data.

For ZeitZeiger[9] (v2.1.3) and TimeSignatR[12] (v1.0), we performed a grid-search to find optimal hyperparameters using the MAE across leave-one-out (LOO) cross-validation as a cost. These two models use an embedded method for automatic feature selection, thus, we did not need to specify the gene features used or perform a wrapper-based feature selection for these models.

We additionally used Taufisher[14] v1.0 first by running LOO cross-validation to determine the optimal number of principal-components (PCs). Since Taufisher lacks an embedded feature selection, we followed the author's recommendations of manually identifying circadian biomarkers. This included selecting the top 10 rhythmic genes based on JTK_CYCLE rhythmicity $Q$-values determined using the *Romanowski* et al.[17] time-course, as well as adding 12 circadian clock genes with significant rhythmicity ($Q < 0.05$).

For PLSR[11], we used the Scikit-learn[71] (v0.24.2) PLSRegression model to develop a multi-output model that predicts the sine and cosine value of the 24-h time, like was used in the ChronoGauge NN-model. Using both a grid-search and recursive feature elimination, we first identified the number of latent factors (ranging from 1 to 10) that gave the minimum fivefold cross-validation MAE. This PLSR model is similar to ChronoGauge in that both output the CT as sine and cosine values, they are Python based, and they do not rely on an embedded feature selection method. We therefore decided to generate an ensemble of sub-predictors using the same SFS wrapper used for ChronoGauge's NN. We ran the SFS algorithm using the same method described previously for the NN to generate an ensemble of 100 sub-predictors, each trained using unique feature sets. Like with ChronoGauge's ensemble, we aggregated sub-predictor outputs using a circular mean.

We compared the accuracy of the ChronoGauge ensemble with each of the models across three test sets (RNA-seq, ATH1, and Ara-Gene) based on the MdAE. The reason the MdAE was used over MAE was because the absolute errors in each set were not normally distributed. We note that this comparison was not part of model selection, as we had already chosen the ChronoGauge ensemble for testing hypotheses. The purpose of the benchmark was to see whether ChronoGauge was competitive with the current state-of-the-art in making CT estimates across unseen control/WT samples.

### Gene-ontology analysis of gene features

For GO term enrichment across all features within the ensemble, we selected unique gene features and used TopGO[38] (v2.54) to find associated biological processes within the org.At.tair.db database[72] (v3.18). We then applied Fisher's test to find associated processes that were more enriched compared to the expected value, using all rhythmic circadian genes as a background. Following Benjami–Hochberg adjustment, we used a threshold of $Q < 0.05$ to select significantly enriched processes. When testing for enrichment, we used all significantly rhythmic genes (meta2d $Q < 0.05$) in *Romanowski* et al.[18] as a background. The justification for this is that we are searching for processes that are enriched above what would be expected across the overall circadian transcriptome.

For the GO term enrichment of different sub-predictors within specific samples, we defined 3 groups of sub-predictors based on their errors being accurate, advanced and delayed. For mutant samples, we first identified sub-predictors that were accurate (absolute error <60 min) in their corresponding WT. Using only these sub-predictors that were accurate in the WT, we identified accurate, advanced and delayed groups of sub-predictors in the mutant sample. In the *Graf* et al.[28] dataset, we selected advanced and delayed groups based on a heuristic threshold determined by the overall magnitude of errors. For plants in the Col-0 background, sub-predictors were defined as advanced based on an error > 60 min and delayed < −60 min, due to their relatively low error. Plants in the Ws-2 background included advanced sub-predictors > 90 min and delayed with < −90 min due to the relatively higher errors. For each group, we identified the gene features used across the sub-predictors and performed GO term analysis as previously described. This allowed us to compare differences in biological process enrichment between accurate and inaccurate sub-predictors for each mutant.

### ChronoGauge application to different species
For applying ChronoGauge's ensemble to species other than *A. thaliana*, we first identified putative orthologs between our gene features and the new species using EnsemblPlants Biomart[73]. If an *A. thaliana* gene mapped to multiple orthologs in a non-model species, then the TPM expression values were averaged to give a single corresponding feature. If an *Arabidopsis* gene feature did not map to any orthologs, then the gene was removed from the feature lists. Sub-predictors were then trained to the original *A. thaliana* training data using the new feature lists, thus we generated an ensemble model for each of the 4 non-model species (*A. halleri*[51,52], *G. max*[15], *B. rapa*[53], and *T. aestivum*[23]). These models were applied to validation data across their respective species. Time-course data for each species were standardized independently from the training using z-score scaling to ensure correspondence in gene expression across species.

### Genome-wide-association-study of circadian temperature response
We acquired genotype information of 153 Swedish ecotypes present within the 1001 Genomes Project database[74] that intersected with the RNA-seq accessions published by *Dubin* et al.[56], in which CT estimates were generated. BCFTOOLs[75] (v1.12) was used to remove indel sites as well as to filter out SNP sites with missing allele frequencies of > 10% and minor allele frequencies <5%, giving sites $N = 880,417$. The remaining missing SNP values were imputed using Beagle[76] (v5.4). Three phenotypes were defined based on the CT estimates for RNA-seq accessions at 10 °C growth, 16 °C growth and the difference between the estimates for each growth group (10 °C–16 °C). GAPIT[77] (v3.4) was run using the model BLINK[78] using population structure covariates based on the first 3 PCs. This was run for each phenotype (CT estimates at 10 °C, CT estimates 16 °C and the difference between these groups). Significant MTAs were selected using a significance threshold of $P < 5.7e\text{-}08$ (corresponding to a Bonferroni adjusted $P < 0.05$).

### Statistical analyses
When looking at differences between groups, we used an independent or paired *T*-test to compare group means where normality assumptions were met based on the results of a Shapiro–Wilks test. Where normality was not assumed, we used either a Mann–Whitney-*U* test to compare independent groups or a Wilcoxon signed-rank test for paired groups. To reduce family wise errors across multiple hypothesis tests, we used Bonferroni adjusted *P*-values to determine significance, except for *P*-values in MetaCycle rhythmicity and GO term enrichment analysis, where *Q*-values were instead generated using the Benjamini–Hochberg adjustment.

## Data availability
All published datasets used in this work are accessible. Accessions for *Arabidopsis thaliana* training data are listed in Supplementary Data 1 and include: E-MTAB-7933[18] [https://www.ebi.ac.uk/biostudies/ArrayExpress/studies/E-MTAB-7933?query=E-MTAB-7933], GSE137732[19] [https://www.ncbi.nlm.nih.gov/geo/query/acc.cgi?acc=GSE137732] & GSE115583[20] [https://www.ncbi.nlm.nih.gov/geo/query/acc.cgi?acc=GSE115583]. Accessions for *A. thaliana* data used in benchmarking are described in Supplementary Data 3 and include the following for RNA-seq experiments: GSE43865[24] [https://www.ncbi.nlm.nih.gov/geo/query/acc.cgi?acc=GSE43865], GSE51578[25] [https://www.ncbi.nlm.nih.gov/geo/query/acc.cgi?acc=GSE51578], SRP064782[26] [https://trace.ncbi.nlm.nih.gov/Traces/?view=study&acc=SRP064782], PRJNA384110[27] [http://ncbi.nlm.nih.gov/bioproject/PRJNA384110], SRP082192[28] [https://trace.ncbi.nlm.nih.gov/Traces/?view=study&acc=SRP082192] & PRJEB10930[29] [https://www.ebi.ac.uk/ena/browser/view/PRJEB10930], and for micro-array experiments: E-GEOD-5612[30] [https://www.ebi.ac.uk/biostudies/arrayexpress/studies/E-GEOD-5612], GSE8365[31] [https://www.ncbi.nlm.nih.gov/bioproject/?term=GSE8365], E-MEXP-1299[32] [https://www.ebi.ac.uk/biostudies/arrayexpress/studies/E-MEXP-1299], E-MEXP-2526[33] [https://www.ebi.ac.uk/biostudies/ArrayExpress/studies/E-MEXP-2526] & GSE50438[34] [https://www.ncbi.nlm.nih.gov/geo/query/acc.cgi?acc=GSE50438]. Accessions not previously listed for *A. thaliana* data that were used in conditional hypothesis testing are described in Supplementary Data 4 and include: GSE116004[50] [https://www.ncbi.nlm.nih.gov/geo/query/acc.cgi?acc=GSE116004] & GSE54680[56] [https://www.ncbi.nlm.nih.gov/geo/query/acc.cgi?acc=GSE54680]. Accessions for non-model species data are described in Supplementary Data 5. This includes:GSE94228[15] [https://www.ncbi.nlm.nih.gov/geo/query/acc.cgi?acc=GSE94228] for *Glycine max* (soybean), GSE123654[53] [https://www.ncbi.nlm.nih.gov/geo/query/acc.cgi?acc=GSE123654] for *Brassica rapa* & DRA005871[51] [https://trace.ncbi.nlm.nih.gov/Traces/?view=study&acc=DRP004575] and DRP005538[52] [https://trace.ncbi.nlm.nih.gov/Traces/?view=study&acc=DRP005538] for *Arabidopsis halleri*. Raw fastq files for *Triticum aestivum* (wheat) are available from: https://opendata.earlham.ac.uk/opendata/data/wheat_circadian_Rees_2021[23]. Source data for figures is provided as a Source Data file. Remaining relevant data and results used can be accessed and downloaded as a zip file in https://osf.io/839em/[79]. Source data are provided with this paper.

## Code availability
ChronoGauge v1.0.0 is available as an open-source software package at https://github.com/ConnorReynoldsUK/ChronoGauge[80], including scripts for generating feature sets, as well as training and testing models in *Arabidopsis*. An additional package is available at https://github.com/ConnorReynoldsUK/ChronoGauge_Xspecies[81], which includes scripts for training and testing ChronoGauge across non-model species. In the manuscript, ChronoGauge was run in Alma Linux v5.14.0 using Python v3.9.5 with Numpy v1.19.5, Pandas v1.3.3, Tensorflow v2.6.0, Scikit-learn v0.24.2, and tqdm v4.61.2. ChronoGauge currently runs using Python v.3.9.5 with Numpy v1.26.4, Pandas v2.2.2, Tensorflow v2.10.0, Scikit-learn v1.5.1, and tqdm v4.66.5. Environments for both versions are available in each of the repositories.

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

## Acknowledgements

We thank Dr Tomoaki Muranaka for his support in preparing *A. halleri* data. We thank Dr Benedict Coombes and Dr Benjamen White for advice on processing genotype data. We thank Dr Azam Lashkari for discussions on our analyses. We thank all members of the Plant Genomics Group at the Earlham Institute and the Dodd Group at the John Innes Centre for discussions on our work. We thank the support delivered by the Research Computing Groups, who deliver and maintain High Performance Computing at the Earlham Institute. We thank all authors who generated the transcriptome accessions used in this study (listed in Supplementary Data 1, 3–5) and made these data available as invaluable and publicly accessible resources. C.R. was supported by the BBSRC-funded Norwich Research Park Biosciences Doctoral Training Partnership grant BB/M011216/1. R.R.P. and A.H. were supported by the Biotechnology and Biological Sciences Research Council (BBSRC), part of UK Research and Innovation; Earlham Institute Strategic Programme Grant BBX011089/1 and BBS/E/ER/230002B (Decode WP2 Genome Enabled Analysis of Diversity to Identify Gene Function, Biosynthetic Pathways And Variation In Agri/Aquacultural Traits). A.D. was supported by BBSRC Institute Strategic Programmes GEN BB/P013511/1 and BRiC BB/X01102X/1.

## Author contributions

C.R. designed and implemented the final model, acquired and processed the majority of the publicly available transcriptome datasets, and performed all downstream analyses. J.C. designed an initial version of the model and supported C.R. in further development and implementation. E.K. processed genotype data and performed the genome-wide-association-study (GWAS). R.R.P. processed some of the publicly available data. H.R., H.K., A.D., and A.H. oversaw and provided biological insights for the project. C.R., H.R., A.D., and A.H. wrote the manuscript.

## Competing interests

J.C. is CEO and co-founder of TraitSeq Ltd. A.H. is Scientific Advisor and co-founder of TraitSeq Ltd. The company played no role in the conception, development or execution of this work. The remaining authors declare no competing interests.
