## [Transparent Peer Review file · Nature Communications]

Machine learning models highlight environmental and genetic factors associated with the Arabidopsis circadian clock

Corresponding Author: Professor Anthony Hall

Version 0:

Reviewer comments:

Reviewer #1

(Remarks to the Author)

This study presents an ensemble model to predict the circadian clock using expression data in Arabidopsis. The work is well-presented, and the conclusions are well-supported. I have only a few minor comments:

- Lines 33-35 require a reference.
- To train the model, only data from Col-0 was used. What are the limitations of this approach?
- Predictions for other species are reported as a mean. It would be interesting if the authors could identify the biological reasons behind the variability in predictions within samples of the same species, for example, in the 36 G. max samples.
- Are all the transcriptomes from Col-0 derived from the same tissue? I did not find this information in the manuscript.

(Remarks on code availability)

The GitHub repository is well-documented and includes a step-by-step guide for installation and usage.

Reviewer #2

(Remarks to the Author)

The authors of this manuscript sought to mathematically address the labor-intensive and costly problem of plant circadian transcriptomics. They developed an ensemble model, ChronoGauge, that can reliably estimate plant endogenous circadian rhythms using data from a single time point. The higher accuracy of ChronoGauge offers a major advantage over similar methods developed in other species, including plants. The authors repeatedly demonstrated the reliability and application potential of ChronoGauge through multiple examples. Especially its application in GWAS is very innovative. First of all, I need to state that because the manuscript involves a lot of mathematical knowledge in the experimental methods and the main text, and I am a chronobiologist, I cannot make any judgment on the mathematical part. I think this method is very helpful for the research of Chronobiology, especially plant circadian clock. By using this model, researchers can re-investigate previous omics data and make some new discoveries or obtain some research ideas. More importantly, adopting this method in subsequent experimental design may also help save labor and cost.

I have the following comments on this manuscript:

1. Because this paper involves the intersection of chronobiology and mathematics, it is recommended that the author explain in detail the relationship between CT (Circadian time), ZT (Zeitgeber time) and sampling time in the experimental method, preferably with diagrams and mathematical formulas, which will help more readers understand and be interested.
2. In the article, it was found that different time points of sampling or different environmental cues can perturb the predictive reliability. The authors just give a simpler speculation, and it is suggested to discuss the possible reasons for this in detail in the discussion section.
3. This manuscript requires further consider about what to do about several possible factors that interfere with prediction

accuracy. There is a key factor here that I think needs to be considered or explained. There are a significant number of rhythmically expressed genes whose expression phase shifts significantly between LD to LL or SD to LL. Even if the average expression abundance of the gene does not change throughout the day under different photoperiods or after release to LL, at a particular time of day, transcript abundance may be upregulated, downregulated, or unchanged due to phase shifts. The number of such genes should be large, which may have a big impact on the prediction accuracy, and we hope the authors will answer how to solve the above situation in the ChronoGauge model.

Minors,

1. The ZTL (AT5G57360) in Supplementary Table 1, although it is a clock gene, has been recognized to be arrhythmic or weakly rhythmic at the transcript level. The authors need to explain why this gene was used in the list.

2. Fig. 6b-d is mixed up with Fig. 5 in the main text and needs to be revised.

(Remarks on code availability)

Reviewer #3

(Remarks to the Author)

This manuscript presents ChronoGauge, a supervised ensemble model based on 100 neural network sub-predictors. The model was developed and tested using modern machine learning frameworks, including TensorFlow, and combines programming languages such as R and Python. The development of machine learning models such as this one can significantly contribute to research by generating more precise hypotheses that can subsequently be tested experimentally. ChronoGauge is a promising approach, being the first machine learning model specifically designed for *Arabidopsis thaliana* transcriptomic studies. The authors demonstrate the model's impressive predictive capabilities, achieving the smallest median absolute errors (MdAEs) across RNA-seq and microarray datasets compared to other existing machine learning models.

The manuscript is well-prepared with excellent eye for detail, which is always nice for reviewers as it means we have very few details to point out. There are three general areas where we feel the manuscript can be improved.

1. Conclusions extracted from model predictions: In the 'Chronogauge reveals...' sections, the authors report that relevant conclusions can be obtained from its predictions. However, most of them have been already described by experimental studies before. On one hand, this is a strength, as it confirms the model's validity and consistency with the existing body of knowledge. On the other hand, the results and discussion sections lack sufficient references to prior literature that could further support this alignment. Most cited works are the papers where data is mined from, and the manuscript would benefit from pointing out the primary research that aligns with ChronoGauge findings. In addition, instead of using terms like "ChronoGauge reveals" in the section titles and similar wording in the conclusions, which implies novelty, the authors could emphasize how ChronoGauge predictions "align with" established findings, citing the relevant literature to support those affirmations.

2. Cross-species applications:

a) ChronoGauge demonstrates its potential by predicting circadian time (CT) with high accuracy across RNA-seq and microarray samples, including datasets from non-model species. However, predicting CT in diverse plants remains challenging due to biological diversity and the inherent difficulty of finding accurate orthologs between species. While the model successfully produces non-random CT predictions in other species, the reported errors suggest that it might not yet be useful for practical applications beyond *Arabidopsis thaliana*. Nevertheless, this represents a meaningful step forward in the field. Future versions of the model could benefit from an expanded training dataset, incorporating circadian transcriptomic data from other plants too. While we appreciate that would be beyond the current study, the title ('plant circadian clock') and abstract ('can be further applied [...] in non-model species') we feel is promising more than the paper justifies. We would ask to rephrase, throughout the manuscript, the claims that this is a 'plant' model while it is an *Arabidopsis* model that can be employed to assess data from other species with far less accuracy.

b) In this section, the authors use datasets from species such as *Arabidopsis halleri* (N=383), *Brassica rapa* (N=48), *Glycine max* (N=36), and *Triticum aestivum* (N=36) for testing the cross-species applications of this model. However, it is unclear why these species were prioritized over others, such as *Solanum lycopersicum* (tomato) or *Chlamydomonas reinhardtii* (green algae), for which circadian transcriptomic data is also available. Have other species been tested unsuccessfully, or did you choose these species for other reasons? What reasons were they? Please include this information in the manuscript.

c) Additionally, the relationship between dataset size and prediction accuracy has caught my attention. *Arabidopsis halleri*, which has the largest dataset (N=383), exhibits the worst CT predictions, while *Glycine max* (N=36) shows the best predictions. This discrepancy might suggest that larger datasets, by capturing more details of the circadian system, introduce more complexity and expose more subtle differences, potentially increasing prediction difficulty. Conversely, smaller datasets may simplify the system, resulting in seemingly better predictions. Testing ChronoGauge on datasets of similar size and quality across different species would help address this concern. Perhaps easier and quicker to do would be to cut down the *halleri* dataset to 10%, 20%, 30% etc of its samples and see what MdAE you get? It should be mentioned in the manuscript whether accuracy scales with N (with data to support it, if yes).

3. ChronoGauge impact on the circadian research community:

a) You will have to write a Data and Code availability section in your methods, where you specify where your data and code

can be found. We note ChronoGauge is shared on GitHub, but this is not mentioned and linked in the manuscript as it should be. Similarly, we would like to have all data included, for example in Github, where currently only a subset of the data is deposited.

b) While sharing the code is an essential step, developing a web tool would greatly enhance the impact of ChronoGauge, enabling researchers with limited computational expertise to explore the model independently. This would not only increase its reach but also facilitate further testing and refinement through community feedback. We would like to ask you to seriously consider doing so for your resubmission.

Overall, ChronoGauge represents an exciting contribution to the field of circadian plant biology and with further improvements—such as creating a user-friendly interface—it could become an indispensable tool for researchers.

Details:

The term ZT features quite a lot and we feel it might be difficult to interpret for those outside the field, because it is not explained apart from a short line in the first section of results. Similarly, CT is not really explained well. We feel the results section "Evaluation of ChronoGauge ensemble" should be broken up into a few sections with more informative titles. It currently covers two main figures, 8 supplemental figures, and a couple of tables. All subsequent results sections are also long and the authors should consider separating these to improve readability. I would guess most of your audience are biologists, who might not be very used to reading computer science.

(Remarks on code availability)

They will have to write a Data and Code availability section in the methods, where they specify where the data and code can be found. We note ChronoGauge is shared on GitHub, but this is not mentioned and linked in the manuscript as it should be. Similarly, we would like to have all data included, for example in Github, where currently only a subset of the data is deposited.

Reviewer #4

(Remarks to the Author)

(Remarks on code availability)

Version 1:

Reviewer comments:

Reviewer #1

(Remarks to the Author)

The authors have addressed all my previous comments, and I have no further suggestions

(Remarks on code availability)

Reviewer #2

(Remarks to the Author)

This revision is a good refinement; I have no further comments.

(Remarks on code availability)

Reviewer #3

(Remarks to the Author)

After reviewing this new version of the manuscript, we have no further comments to make. We are satisfied the authors have addressed our initial comments as far as could reasonably be expected.

(Remarks on code availability)

Reviewer #4

(Remarks to the Author)

I co-reviewed this manuscript with one of the reviewers who provided the listed reports. This is part of the Nature

Communications initiative to facilitate training in peer review and to provide appropriate recognition for Early Career Researchers who co-review manuscripts.

(Remarks on code availability)

REVIEWER COMMENTS

We would like to thank the reviewers for their valuable comments and suggestions on our manuscript. Below, we provide detailed responses to all reviewers' individual queries. To make this easy to interpret, we have put **reviewer comments in blue bold**, our response in standard text and *excerpts from the manuscript in red italics*. All references to the manuscript are associated with line numbers (based on Microsoft Word 365 format).

Reviewer #1 (Remarks to the Author):

This study presents an ensemble model to predict the circadian clock using expression data in Arabidopsis. The work is well-presented, and the conclusions are well-supported. I have only a few minor comments:

- Lines 33-35 require a reference.

To support this general description of the plant circadian clock, we have included the review by *McClung (2006)* (10.1105/tpc.106.040980) as a reference in Line 34.

- To train the model, only data from Col-0 was used. What are the limitations of this approach?

Since the model is fit exclusively to Col-0 expression, it would be intuitive that predictions outside of Col-0 accessions should be less accurate (since many circadian genes will almost certainly be expressed differently). This is a form of overfitting, but it is currently unavoidable since there are few time-series datasets outside of Col-0.

An example is included in Figure 4, where we compare the test predictions of Col-0 and Ws-2 accessions. It was not remarkable that Col-0 gave more accurate CT predictions than Ws-2 by itself, since we would expect this because of the overfitting to Col-0. We however felt it was interesting that at both time-points, Ws-2 gave significantly advanced predictions compared to Col-0, which is consistent with previous studies that show a shorter period length in Ws-2. This is now explained in the Results section (Line 303-307):

“We note the Ws-2 samples were significantly advanced ($P < 0.01$) from Col-0 at each timepoint (Fig. 4c, f). These advanced estimates likely reflect differences in circadian transcriptional dynamics between the two accessions, since ChronoGauge was trained only using Col-0 and not Ws-2, and also coincides with previous analyses that have shown Ws-2 plants display a marginally shorter period length than Col-0 plants (by ~30 mins) under LL⁴⁹”

- Predictions for other species are reported as a mean. It would be interesting if the authors could identify the biological reasons behind the variability in

predictions within samples of the same species, for example, in the 36 *G. max* samples.

We did not place an emphasis on identifying specific biological reasons for variation across time-points in non-model species, as there is the potential for artefacts being introduced as a result of uncertain or duplicated ortholog mappings. Except for *A. halleri*, the error and variation of CT estimates between different time-points is also not that different (Supplementary Fig. 16).

Using only *G. max* as an example (as this displayed the most accurate results compared with other species), we attempted to identify biological processes that were enriched in sub-predictors that were accurate (< 60 mins absolute error), advanced (> 90 mins error) and delayed (< -90 mins error) for each time-point using the fingerprint method described in Figure 3:

We see that there is an enrichment in “glucosinolate biosynthetic process” at ZT0 and photosynthetic processes between ZT12-20 in advanced groups compared with accurate groups, suggesting that dysregulation in genes associated with these pathways may be responsible for an advanced phase (relative to *A. thaliana*) at these points. Likewise, a subtle enrichment of photosynthetic processes is enriched at ZT8 in the delayed group, suggesting dysregulation in genes under this pathway could be responsible for the delayed predictions at this time-point.

We are not overly confident in our findings here and believe the potential for errors being introduced as a result of ortholog mapping between species means that the analysis is not robust enough. Therefore, we decided to not include the above analysis in our manuscript.

- Are all the transcriptomes from Col-0 derived from the same tissue? I did not find this information in the manuscript.

The tissues for each experiment are listed in Supplementary data 1-4. However, since this was not clear from the text, we have included descriptions of the tissue used in each dataset throughout the text.

Reviewer #1 (Remarks on code availability):

The GitHub repository is well-documented and includes a step-by-step guide for installation and usage.

Reviewer #2 (Remarks to the Author):

The authors of this manuscript sought to mathematically address the labor-intensive and costly problem of plant circadian transcriptomics. They developed an ensemble model, ChronoGauge, that can reliably estimate plant endogenous circadian rhythms using data from a single time point. The higher accuracy of ChronoGauge offers a major advantage over similar methods developed in other species, including plants. The authors repeatedly demonstrated the reliability and application potential of ChronoGauge through multiple examples. Especially its application in GWAS is very innovative. First of all, I need to state that because the manuscript involves a lot of mathematical knowledge in the experimental methods and the main text, and I am a chronobiologist, I cannot make any judgment on the mathematical part. I think this method is very helpful for the research of Chronobiology, especially plant circadian clock. By using this model, researchers can re-investigate previous omics data and make some new discoveries or obtain some research ideas. More importantly, adopting this method in subsequent experimental design may also help save labor and cost.

I have the following comments on this manuscript:

1. Because this paper involves the intersection of chronobiology and mathematics, it is recommended that the author explain in detail the relationship between CT

(Circadian time), ZT (Zeitgeber time) and sampling time in the experimental method, preferably with diagrams and mathematical formulas, which will help more readers understand and be interested.

We understand that the current use of sampling-time, ZT, CT and predicted/estimated CT is largely unexplained to readers. We apologise for not explaining this chronobiology-specific terminology. We have attempted to explain this further in the text (Line 45-54):

“The CT refers to the internal time of an organism irrespective of external cues, and is often used to label the time-of-sampling under free running conditions (continuous light (LL) in plants). This contrasts with the “zeitgeber time” (ZT), which refers to the experimental time relative to environmental cues, and is used to label the experimental time under a light-dark (LD) cycle. These predictive models are fit to labeled time-course gene expression data, which can then be used to estimate an organism’s internal time in single time-pointed datasets. Time estimates in previous models trained on LL-condition data appear to correspond with both CT and ZT labels without using any mathematical adjustment⁸⁻¹⁴, suggesting that circadian variation can be captured even when samples are harvested under a LD cycle.”

We believe this addition suffices, and an additional figure may be unnecessary or could potentially cause more confusion to readers.

We note that we do not use any mathematical adjustment to CT labels or predictions based on period length. We assume that sampling time, ZT and CT broadly correspond to one another - we show that this is largely the case in our test results, which generally includes acceptable accuracies in samples harvested either under constant-light conditions or light-dark cycles. **This is consistent with previous CT estimation models, which also do not appear to adjust the CT labels/predictions.**

2. In the article, it was found that different time points of sampling or different environmental cues can perturb the predictive reliability. The authors just give a simpler speculation, and it is suggested to discuss the possible reasons for this in detail in the discussion section.

Our key aims in developing ChronoGauge were to generate testable hypotheses about the interaction between the environment and the clock. These are only hypotheses and will need to be tested further to validate or reject them. At this point, we would not like to speculate further on more fundamental and molecular reasons for our findings until our hypotheses have been tested further through experimental analyses.

With this in mind, we believe our more broad suggestions for the reasons causing the perturbations we observe (in which we refer to

what has been observed in previous experiments with similar experimental conditions, where possible) are sufficient.

3. This manuscript requires further consider about what to do about several possible factors that interfere with prediction accuracy. There is a key factor here that I think needs to be considered or explained. There are a significant number of rhythmically expressed genes whose expression phase shifts significantly between LD to LL or SD to LL. Even if the average expression abundance of the gene does not change throughout the day under different photoperiods or after release to LL, at a particular time of day, transcript abundance may be upregulated, downregulated, or unchanged due to phase shifts. The number of such genes should be large, which may have a big impact on the prediction accuracy, and we hope the authors will answer how to solve the above situation in the ChronoGauge model.

ChronoGauge's sequential feature selection approach means that genes which undergo a significant phase shift are unlikely to be selected as features for CT prediction, at least for the conditions observed in the training data (both continuous light and diurnal following 12:12 and long-day/16:8 entrainment). This is supported in Supplementary Fig. 11, which shows the expression patterns of the top selected gene features are quite robust across experiments with different entrainment/sampling photoperiods.

As a neural-network model, ChronoGauge makes predictions based on complex and non-linear interactions between gene features. This means that even if some genes do have a phase shift between photoperiods, the CT can still be accurately estimated assuming collective patterns in gene expression broadly align with what was observed in the training data. We briefly refer to the neural-network's ability to make complex transformations in the Discussion (Line 455-458):

“It is possible that the expression data collectively contains important and complex information that linear-based models are less able to exploit compared to a NN-transformation”

Moreover, our ensemble approach is not reliant on a small set of genes but instead a set of sub-predictors. While the accuracy of sub-predictors can vary due to variation in the expression of some genes, the circular-mean of predictions across all sub-predictors tend to have an error close to zero in WT/control samples. This ensemble method ensures predictions are robust, even when individual gene features vary.

We did not include a time-series entrained under short-day/8:16 conditions in the training data, thus cannot guarantee that genes will not undergo a phase shift in this condition compared with neutral-, long-day or LL conditions. Looking at a breakdown of predictions made per experiment in the test datasets, we do observe that predictions obtained from experiments entrained/sampled under 8:16 (short-day) conditions tend to be less accurate than other conditions:

Experiment	Method	Entrainment photoperiod	Sampling photoperiod	MdAE (mins)	MAE (mins)
Rugnone et al. (N = 18)	RNA-seq	16:8	16:8	15.3 (± 16.5)	21.2 (± 16.5)
Miller et al. (N = 3)	RNA-seq	16:8	16:8	66.9 (± 57.5)	101.9 (± 57.5)
Takahashi et al. (N = 12)	RNA-seq	12:12	LL	72.6 (± 39.9)	59.1 (± 39.9)
Ezer et al. (N = 8)	RNA-seq	8:16	8:16	111.8 (± 58.8)	108.0 (± 58.8)
Graf et al. (N = 6)	RNA-seq	12:12	12:12	9.4 (± 4.2)	11.4 (± 4.2)
Dubois et al. (N = 12)	RNA-seq	16:8	16:8	16.3 (± 15.0)	20.4 (± 15.0)
Edwards et al. (N = 13)	Microarray ATH1	12:12	LL	77.6 (± 57.8)	80.1 (± 57.8)
Covington et al. (N = 12)	Microarray ATH1	12:12	LL	36.4 (± 60.5)	66.0 (± 60.5)
Michael et al. (Col-0, N = 6)	Microarray ATH1	16:8	16:8	34.1 (± 21.8)	40.2 (± 21.8)
Michael et al. (Col-0, N = 6)	Microarray ATH1	8:16	8:16	79.1 (± 48.8)	93.4 (± 48.8)
Michael et al. (Ler, N = 6)	Microarray ATH1	8:16	8:16	100.2 (± 63.7)	100.9 (± 63.7)
Espinoza et al. (N = 15)	Microarray ATH1	16:8	LL	43.3 (± 32.5)	45.4 (± 32.5)
Espinoza et al. (N = 15)	Microarray ATH1	16:8	16:8	43.0 (± 18.0)	39.3 (± 18.0)
Endo et al. (N = 36)	Microarray AraGene	16:8	16:8	49.4 (± 56.0)	70.0 (± 56.0)
Endo et al. (N = 36)	Microarray AraGene	8:16	8:16	104.0 (± 73.2)	109.4 (± 73.2)

We have added the table to Supplementary Table 6 and have additionally added the following text highlighting these limitations in the results section (Line 194-204):

“Our training data did not include a time-course experiment entrained to short-day conditions (only neutral- or long-day), thus it was unclear whether ChronoGauge could accurately estimate the CT with said short-day entrainment. Looking at model performance within each experiment (Supplementary Table 6), we do see that experiments entrained and harvested under short-day conditions had higher MdAEs (though not unacceptable, using 120 mins as a threshold) compared with all other conditions. This suggests overfitting has occurred, which on one hand may highlight differences in clock dynamics between plants entrained under short-day and neutral-/long-day photoperiods that have been observed previously³⁶. On the other hand, it also suggests ChronoGauge may be less reliable when applied to samples harvested under short-day photoperiods. Despite the increased error, the fact that CT estimates are not random in the short-day experiments indicates

that there is still some overlap in circadian gene expression compared with other experiments.”

We further comment on this in the Discussion (Line 509-513):

“A potential limitation of ChronoGauge is that it appeared to be overfit to data that included neutral-day or long-day entrainment, meaning applying ChronoGauge may give less reliable results when making comparisons of samples entrained under short-day conditions. This limitation could be overcome by including a high quality time-course spanning 2 or more days with short-day entrainment into the training data, but to the best of our knowledge, no such dataset is available.”

It may also be relevant that even under similar experimental conditions (e.g. short-day, long-day, LL experiments), circadian regulated genes do not necessarily exhibit identical phases and rhythmicity metrics. For example, *Brooks et al.* (10.1177/07487304231179600) showed that in the mouse liver, the phase of overall gene expression is not consistent across LL or LD studies, with technical factors appearing to be a strong source of variation across experiments. This may also be the case in plants and could be a strong limitation of conventional circadian analyses. We briefly commented on this in Line 37-42.

Minors,

1. The *ZTL* (AT5G57360) in Supplementary Table 1, although it is a clock gene, has been recognized to be arrhythmic or weakly rhythmic at the transcript level. The authors need to explain why this gene was used in the list.

Based on MetaCycle’s meta2d Q-values as a metric for rhythmicity, *ZTL* expression displayed a significant rhythm ($Q = 0.0005$) in the *Romanowski et al.* LL time-series, with a modest rhythm ($Q = 0.09$) in the *Yang et al.* LL time-series. Upon visual inspection, we do not agree with the suggestion that LL is necessarily arrhythmic or weakly rhythmic under continuous light.

The genes listed in Supplementary Table 1 are those that a biologist might hand pick as circadian regulators based on experimental evidence for said regulation. We picked these genes as they are known regulators of the clock, and the fact they are expressed in all training datasets.

We additionally note that as the neural network model uses non-linear transformations to make a CT prediction, it is not truly required that each gene feature (or even any gene feature) is rhythmic to make an accurate prediction – only that the collective composition of their expression can reflect the sampling time.

2. Fig. 6b-d is mixed up with Fig. 5 in the main text and needs to be revised.

Thank you, this has been corrected.

Reviewer #3 (Remarks to the Author):

This manuscript presents ChronoGauge, a supervised ensemble model based on 100 neural network sub-predictors. The model was developed and tested using modern machine learning frameworks, including TensorFlow, and combines programming languages such as R and Python. The development of machine learning models such as this one can significantly contribute to research by generating more precise hypotheses that can subsequently be tested experimentally. ChronoGauge is a promising approach, being the first machine learning model specifically designed for *Arabidopsis thaliana* transcriptomic studies. The authors demonstrate the model's impressive predictive capabilities, achieving the smallest median absolute errors (MdAEs) across RNA-seq and microarray datasets compared to other existing machine learning models.

The manuscript is well-prepared with excellent eye for detail, which is always nice for reviewers as it means we have very few details to point out. There are three general

areas where we feel the manuscript can be improved.

1. Conclusions extracted from model predictions: In the 'Chronogauge reveals...' sections, the authors report that relevant conclusions can be obtained from its predictions. However, most of them have been already described by experimental studies before. On one hand, this is a strength, as it confirms the model's validity and consistency with the existing body of knowledge. On the other hand, the results and discussion sections lack sufficient references to prior literature that could further support this alignment. Most cited works are the papers where data is mined from, and the manuscript would benefit from pointing out the primary research that aligns with ChronoGauge findings. In addition, instead of using terms like "ChronoGauge reveals" in the section titles and similar wording in the conclusions, which implies novelty, the authors could emphasize how ChronoGauge predictions "align with" established findings, citing the relevant literature to support those affirmations.

We accept that the original phrasing could appear as though we were presenting novelties that may not necessarily exist. We have therefore changed all cases of "*reveals*" within the text to "*indicates*" or "*highlights*", as we feel this is less promising of novelty. Where possible, we have expanded discussions surrounding our findings and their alignment (or lack of) with previous work.

2. Cross-species applications:

a) ChronoGauge demonstrates its potential by predicting circadian time (CT) with high accuracy across RNA-seq and microarray samples, including datasets from non-model species. However, predicting CT in diverse plants remains challenging due to biological diversity and the inherent difficulty of finding accurate orthologs between species. While the model successfully produces non-random CT predictions in other species, the reported errors suggest that it might not yet be useful for practical applications beyond *Arabidopsis thaliana*. Nevertheless, this represents a meaningful step forward in the field. Future versions of the model could benefit from an expanded training dataset, incorporating circadian transcriptomic data from other plants too. While we appreciate that would be beyond the current study, the title ('plant circadian clock') and abstract ('can be further applied [...] in non-model species') we feel is promising more than the paper justifies. We would ask to rephrase, throughout the manuscript, the claims that this is a 'plant' model while it is an *Arabidopsis* model that can be employed to assess data from other species with far less accuracy.

To meet the reviewers recommendation of not overpromising ChronoGauge's application in non-model plant species, we have rephrased the text so ChronoGauge is described as a "*CT estimation model for Arabidopsis*" rather than "*plant CT estimation model*".

Referring to ChronoGauge's applications to non-model plant species, we now describe ChronoGauge as being applied to make "*non-random predictions*" in non-model plants.

b) In this section, the authors use datasets from species such as *Arabidopsis halleri* (N=383), *Brassica rapa* (N=48), *Glycine max* (N=36), and *Triticum aestivum* (N=36)

for testing the cross-species applications of this model. However, it is unclear why these species were prioritized over others, such as *Solanum lycopersicum* (tomato) or *Chlamydomonas reinhardtii* (green algae), for which circadian transcriptomic data is also available. Have other species been tested unsuccessfully, or did you choose these species for other reasons? What reasons were they? Please include this information in the manuscript.

In this test of ChronoGauge's applications within non-model plant species, we chose a subset of important crop species (with circadian-controlled time-courses) with two aims: firstly, to demonstrate the utility with closely related species (i.e. within the Brassicaceae) in addition to testing in more diverse species, and secondly, to demonstrate the reliability and transferability of the predictors. We did not aim to be exhaustive in our search for these datasets, this was simply a proof of concept. Datasets that were considered low quality (those that displayed substantial noise and few known circadian-regulated genes) were not included. The *Arabidopsis halleri* data set was included to allow us to test for season variation in clock function (as well being an exceptional opportunity to test the model on plants harvested from the wild). The text has been revised to convey this (Line 345-351):

“We tested this approach by making validation CT estimates on RNA-seq datasets firstly including Arabidopsis halleri samples harvested from the wild^{51,52} (N = 383), as this dataset represents a closely related species to A. thaliana with the potential for comparison across different seasonal conditions. We also tested CT estimation in more distantly related species that contain known clock gene orthologs with time-courses that were sampled under controlled LL conditions including Brassica rapa⁵³ (N = 48), Glycine max¹⁵ (N = 36) and Triticum aestivum²³ (N = 36) (Supplementary Data 4).”

c) Additionally, the relationship between dataset size and prediction accuracy has caught my attention. *Arabidopsis halleri*, which has the largest dataset (N=383), exhibits the worst CT predictions, while *Glycine max* (N=36) shows the best predictions. This discrepancy might suggest that larger datasets, by capturing more details of the circadian system, introduce more complexity and expose more subtle differences, potentially increasing prediction difficulty. Conversely, smaller datasets may simplify the system, resulting in seemingly better predictions. Testing ChronoGauge on datasets of similar size and quality across different species would help address this concern. Perhaps easier and quicker to do would be to cut down the *halleri* dataset to 10%, 20%, 30% etc of its samples and see what MdAE you get? It should be mentioned in the manuscript whether accuracy scales with N (with data to support it, if yes).

The reviewer suggests *Arabidopsis halleri* gives the “worst predictions”, yet based on MdAE, it gave the second lowest error for non-model species after *G. max*. Instead, *B. rapa* gave the highest error at 169.2 mins (**we note that the original Fig. 5a gives an incorrect MdAE value for *B. rapa*. We apologize for this oversight and have revised the figure**).

Perhaps the reviewer was referring to the Pearson correlation coefficient (r) of the predicted CT vs. true sampling time (Shown in Supplemental Fig. 16). Here, we do see that *A. halleri* gives a weaker correlation ($r = 0.882$) compared with all other species ($r > 0.950$), though this analysis approach is more sensitive to extreme outliers (which we expect to see in *A. halleri*, since it was harvested in the wild rather than under controlled conditions), hence why we chose to focus on using MdAE.

Partitioning the data into different fractions indicates that the median-of-MdAEs across 1000 repetitions is similar to the MdAE across the whole dataset ($N = 383$):

MdAE across all *A. herri* samples

While the distribution at fraction 0.1 is highly variable compared with other fractions, this is due to the fact many samples are extreme outliers in this dataset (that have extremely high or low error compared with the overall median/mean).

A. halleri is an exceptional dataset in that it includes only plants harvested from the wild in non-controlled conditions. Many of these

plants are known to be infected with TuMV (*Honjo et al.* 10.1038/s41396-019-0519-4), and may be impacted by other, unknown factors that interfere with circadian regulation of transcription. Additionally, these plants were harvested across 4 different seasons, which contrasts strongly with the other species that were harvested in continuous light under experimentally-controlled temperature. With these parameters in mind, outliers in predictions for the *A. halleri* are to be expected. That we would observe non-random predictions in these wild samples was not at all guaranteed.

In summary, we do not believe the variation in performance between species is associated with the sample sizes (N) used for said species.

3. ChronoGauge impact on the circadian research community:

a) You will have to write a Data and Code availability section in your methods, where you specify where your data and code can be found. We note ChronoGauge is shared on GitHub, but this is not mentioned and linked in the manuscript as it should be. Similarly, we would like to have all data included, for example in Github, where currently only a subset of the data is deposited.

Thank you for noting this oversight. We have added availability sections for both Data (Line 764-770) and Software (Line 771-780).

As the original GitHub repository (<https://github.com/ConnorReynoldsUK/ChronoGauge>) does not currently include data and examples of ChronoGauge's use in non-model species, we have developed a separate repository specifically for cross-species applications (https://github.com/ConnorReynoldsUK/ChronoGauge_Xspecies). We believe a separate repository is appropriate for cross-species analyses to make this very specific and less reliable application of ChronoGauge more understandable and accessible.

The accession IDs for all datasets used are found in Supplementary Data 1-4. Additionally, we deposited relevant data and code used in this work in a zip file that can be downloaded using OSF (<https://osf.io/839em/>). We note that uploading a zip file on OSF for data accessibility has been used in Nature Communications publications before (by *Cheng et al.*, 10.1038/s41467-021-25893-w).

b) While sharing the code is an essential step, developing a web tool would greatly enhance the impact of ChronoGauge, enabling researchers with limited

computational expertise to explore the model independently. This would not only increase its reach but also facilitate further testing and refinement through community feedback. We would like to ask you to seriously consider doing so for your resubmission.

We appreciate the reviewer's suggestions on releasing a web tool and understand that this could make ChronoGauge more accessible to researchers who lack computational expertise. However, this ultimately falls outside the scope of our work. We chose to publish ChronoGauge and its findings in Nature Communications to promote innovation in machine learning within circadian biology, and have presented our method as an accessible and open-source software. We also have provided Jupyter Notebooks in ChronoGauge's main repository (<https://github.com/ConnorReynoldsUK/ChronoGauge>), which we believe many circadian biologists will be able to follow.

We additionally note that maintaining and hosting a web tool is highly resource-intensive and demands expertise in software development. This is beyond our expertise and resources. We would however welcome future collaborations with members of the community who possess expertise in software development to implement such a web tool.

Overall, ChronoGauge represents an exciting contribution to the field of circadian plant biology and with further improvements—such as creating a user-friendly interface—it could become an indispensable tool for researchers.

Details:

The term ZT features quite a lot and we feel it might be difficult to interpret for those outside the field, because it is not explained apart from a short line in the first section of results. Similarly, CT is not really explained well.

We feel the results section "Evaluation of ChronoGauge ensemble" should be broken up into a few sections with more informative titles. It currently covers two main figures, 8 supplemental figures, and a couple of tables. All subsequent results sections are also long and the authors should consider separating these to improve readability. I would guess most of your audience are biologists, who might not be very used to reading computer science.

We thank the reviewer for the suggestion to divide this section to make the text more clear. We have added further explanation in the text for the terms "ZT" and "CT" as described previously (Line 45-54). We have split the "*Evaluation of ChronoGauge ensemble*" section into two: one about the evaluation of ChronoGauge's predictive performance (Line 142 "*Evaluation of ChronoGauge ensemble's predictive accuracy*"), and another about model interpretation (Line 205, "*Interpretability of ChronoGauge ensemble*").

The large number of supplemental figures referenced by ChronoGauge's predictive performance section is essential so we can be as open as possible about the accuracy of ChronoGauge compared with other CT estimation models. Without tables listing each metric (MdAE, MAE and r) and scatter/box plots to show the distribution of errors and predictions, the results of the benchmark would not be complete and specialist readers may not be satisfied with our claims regarding ChronoGauge's superior predictive performance in this work.

Reviewer #3 (Remarks on code availability):

They will have to write a Data and Code availability section in the methods, where they specify where the data and code can be found. We note ChronoGauge is shared on GitHub, but this is not mentioned and linked in the manuscript as it should be. Similarly, we would like to have all data included, for example in Github, where currently only a subset of the data is deposited.

We believe our Data and Software availability statements (discussed previously) now resolve this issue.

Reviewer #4 (Remarks to the Author):

We note that the following figures have been adjusted:

Figure 5a: the MdAE for *Brassica rapa* has been adjusted from the incorrect value of 141.5 mins to the correct value of 169.2 mins.

Supplementary Table 6 has been added to give a full breakdown of ChronoGauge's predictive performance across individual experiments within the test datasets.